# Epidemiological analysis reveals coral species affected by stony coral tissue loss disease present a similar epizootic progression despite differences in susceptibility and population impact

Edgar Omar Guzmán-Urieta[1,2]*, Eric Jordán-Dahlgren[3], Lorenzo Alvarez-Filip[2]

**1** Posgrado en Ciencias del Mar y Limnología, Universidad Nacional Autónoma de México; Av. Universidad 3000, Ciudad Universitaria Coyoacán, C.P. 04510, Ciudad de México, México **2** Biodiversity and Reef Conservation Laboratory, Unidad Académica de Sistemas Arrecifales, Instituto de Ciencias del Mar y Limnología, Universidad Nacional Autónoma de México, Quintana Roo, México, **3** Laboratorio de Ecología Arrecifal, Unidad Académica de Sistemas Arrecifales, Instituto de Ciencias del Mar y Limnología, Universidad Nacional Autónoma de México, Quintana Roo, México

\* omarzo@comunidad.unam.mx

## Abstract

Stony coral tissue loss disease (SCTLD) is one of the most aggressive coral syndromes recorded, affecting over 30 scleractinian species and causing high mortality rates. Despite its impact, most available information is derived from assessments that estimate prevalence at a single point in time, rather than examining its temporal dynamics. This study analyzed the susceptibility of 16 coral species to SCTLD and tracked the 2018–2019 outbreak on a fringing reef in the Mexican Caribbean using epidemiological methods commonly employed in human epidemics but rarely in coral epizootics. Between June 2018 and July 2019, we monitored 990 coral colonies. For each affected colony, we estimated the progression of tissue death over time, allowing us to identify the days of lesion onset and total tissue mortality. To assess vulnerability and provide a detailed prognosis of outbreak progression, magnitude, and severity during the epizootic we employed epidemic curves, Kaplan-Meier risk and survival functions, and measures of period prevalence and mortality. Epidemiological parameters from these analyses were integrated into a multi-dimensional framework for a comprehensive assessment of coral susceptibility. Our findings revealed that species-specific susceptibility was associated with the risk, magnitude, and severity of the epizootic but not with its progression. Temporal analyses revealed community-level patterns, including secondary outbreak waves following increases in mortality. This suggests a potential feedback mechanism, where mortality may contribute to secondary transmission events, a phenomenon not previously described in the study of coral epizootics. As a contribution to the characterization of coral susceptibility to SCTLD, we outline a four-level framework based on diverse epidemiological

**Data availability statement:** All relevant data are within the paper and its Supporting Information files.

**Funding:** All the sources of support received during this study were: The Institute of Marine Sciences and Limnology (ICMyL) and the National Autonomous University of Mexico (UNAM) providing financial and logistic support to EJ-D; Mexican Council of Science and Technology (CONACYT) grant the doctoral scholarship granted to EOG-U (895565), and a research grant awarded to LA-F (FORDECYT-PRONACES/425888/2020). ICMyL: https://www.icmyl.unam.mx/ UNAM: https://www.unam.mx/ CONACYT: https://secihti.mx/becas_posgrados/ There was no additional external funding received for this study.

**Competing interests:** The authors have declared that no competing interests exist.

indicators, beyond prevalence. This approach revealed a higher-than-expected susceptibility in *Siderastrea siderea*, *Agaricia agaricites* and *Agaricia tenuifolia,* compared to previous studies. This study underscores the importance of epidemiological approaches in investigating coral epizootics. By challenging traditional reliance on prevalence measurements, our findings offer a novel perspective on coral disease dynamics.

## Introduction

Coral reefs are one of the most biodiverse ecosystems in the world [1]; their complex structure and energetic dynamism allow the development of hundreds of species and provide goods and services for coastal human communities [2]. Paradoxically, the entire system depends on the tiny and highly vulnerable polyps of hermatypic corals. These organisms have narrow environmental requirements, and repeated or prolonged deviations from these conditions can lead to physiological dysfunctions and increase their susceptibility to pathogens [3]. Therefore, it is not surprising that the occurrence of coral disease and its severity has rapidly increased during the last decades following an increase in environmental pressures in the form of water pollution, eutrophication and ocean warming [4,5].

Among coral disease outbreaks, the most devastating for the Caribbean reefs in the past decade has been caused by a white syndrome known as stony coral tissue loss disease (SCTLD) [6–8]. SCTLD is a breakpoint for the ecological integrity of Caribbean reefs, as this disease drastically changed coral cover and coral composition and impacted key processes such as habitat provisioning and production of calcium-carbonated structures [9–12]. Despite the severity of this coral syndrome, the etiology of SCTLD remains unknown. Multiple bacteria and some viruses have been linked to lesions (e.g., [13–18]) yet their role in the development of the disease is unclear, whether as primary or secondary pathogens [19–21]. At the regional scale, the spread corresponds to a contagious pattern [22], but at the local level the pattern is inconsistent. On the one hand, Williams et al. [23] and Dahlgren et al. [24] observed spatial dependence among SCTLD colonies and a higher risk of lesions occurrence in colonies near diseased neighbors, on reefs in the Lower Florida Keys and Bahamas, respectively. On the other hand, Sharp et al. [25] and Guzmán-Urieta & Jordán-Dahlgren [26] observed that SCTLD spatial incidence was independent of nearest neighbor condition. This suggests that at a small scale, the spread pattern could be influenced by environmental characteristics, host conditions, and species-specific susceptibility [23,26,27].

SCTLD has severely impacted populations of over 30 coral species [6]. The most highly affected are now at a clear risk of extinction across their distribution ranges [28]. Other susceptible species, such as brain corals, which experienced comparatively lower declines (30–70%) may have compromised their viable population sizes [26,29]. Among the coral species affected by SCTLD, there is high variability of epidemiological indicators such as the time to disease onset, lesion progression rate,

and prevalence that was interpreted as species-specific susceptibility [7,8,27,29–33]. Yet, prevalence, i.e., the proportion of diseased individuals in a specific place and time [34], is the primary epidemiological descriptor used to quantify the degree of susceptibility of different species (S1 Table) and to compare the disease magnitude and severity between sites. For example, in the first published SCTLD case definition [30], species susceptibility was categorized by its prevalence, tissue mortality rates, and sequence of lesion appearance; likewise, many other studies have assigned level of susceptibility, described regional trends and compared between localities or environments using prevalence (e.g., [32,35]).

However, prevalence depends on the moment of measurement [36], and it does not allow for the distinction of epizootic progression, offering a limited perspective. For example, suppose prevalence is measured at different times within the same population impacted by an epizootic event (Fig 1A). In that case, different prevalence values can be obtained depending on the measurement time, likely obscuring the understanding of the disease outbreak. A relevant example is the study by Dahlgren et al. [24], which documented spatial and temporal variation in SCTLD prevalence across coral species over a 6–10 month period in Bahamian reefs. This underscores the challenge of comparing species susceptibility based solely on prevalence, as each species may exhibit different prevalence levels depending on the stage of the outbreak (Fig 1B). In contrast, tracking the event over time allows for the identification of epidemiologically and ecologically meaningful patterns in species susceptibility.

A classic perspective for analyzing the progression of an epidemic is the use of epidemic curves (epi-curves). Since its implementation by John Graunt in the middle of the 17th century with the Plague [37], epi-curves have been commonly used to analyze infectious and non-infectious disease outbreaks in humans and animals (i.e., [38,39]). These curves show the onset of cases (incidence) of an outbreak. They help to determine the progression, development and magnitude of an epidemic or epizootic, distinguish epizootic from enzootic phases, and enabling comparisons with potentially correlated factors. The shape of an epi-curve can inform about mode of spread, which could be from a common source (with punctual, continuous or intermittent discharges) or a propagated outbreak due to transmission from one individual to another.

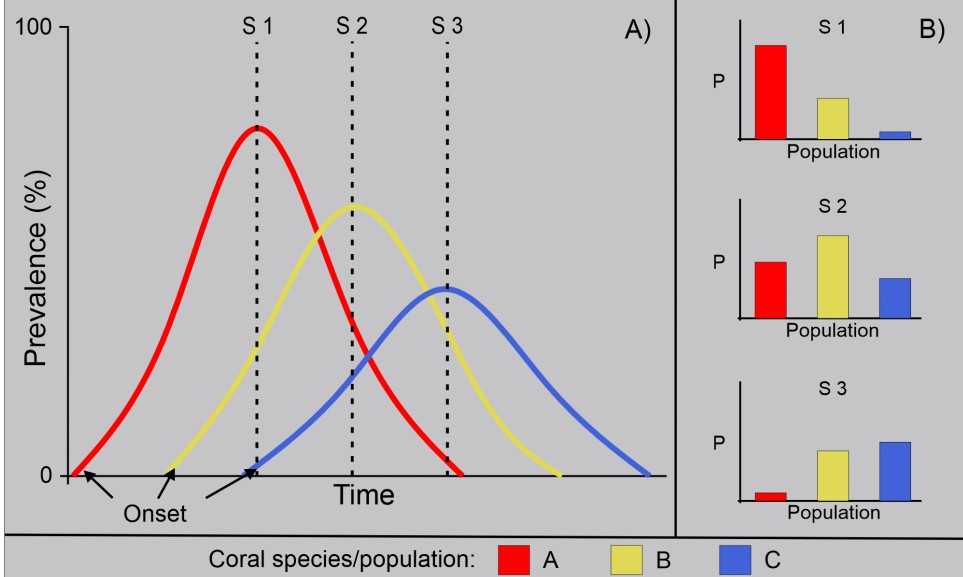

**Fig 1. Schematic of the SCTLD prevalence changes through time.** A) Differentiating among the three theoretical responses of coral species susceptibility to SCTLD: high (red), intermediate (yellow) and low (blue). Black arrows mark the epizootic start of each species based on the observations of FDEP [30]. From this differential epizootic start a prevalence peak is inferred at different times. Dashed lines show three hypothetical sampling periods (S1, S2 & S3) that produce the different prevalence estimates shown in B). B) Bars indicate that species prevalence will vary depending on the sampling time.

Also, if epi-curves are normalized as incidence rate curves, they can be used to compare the severity of the epizootic magnitude between sites and/or species [34,40–43]. Characteristics of epi-curves such as peak timing, peak height, curve duration, and curve shape, are all relevant measures of epizootic severity and are helpful to predict future events [41,42]. And, assuming exposure to an agent at the same time, comparison of epi-curves can help determine the degree of susceptibility between sites or/and species.

Despite the great level of detail they provide, epi-curves have not been used much on coral epizootics. We systematically searched for studies on coral epizootics in Web of Science [44] (S2 Table) and found that, among 439 studies conducted between 1995 and 2025, only 3.4% used or implemented approximations of epi-curves in their analysis, with only two studies focusing on SCTLD. Other 15.9% used prevalence *vs.* time curves, which show changes in the diseased population and can differentiate between enzootic and epizootic phases [45], yet do not portray accurately the epizootic progression as the curve is shaped by new cases, deaths, recoveries and not found individuals (censoring). And the 80.7% remaining only uses the words prevalence/incidence without epizootic analysis or use prevalence measures to compare affectation between sites or species. The implementation of epi-curves in SCTLD analysis would therefore allow for further understanding of the epizootic progression and an opportunity to improve the current epizootic surveillance protocols, allowing for better predictive capacity.

In this study, we aimed to investigate the progression, magnitude and severity of an SCLTD epizootic by tracking the condition of nearly one thousand colonies from 16 coral species for thirteen months in a shallow fringing reef. We reconstructed individual tissue loss trajectories for each affected colony, to estimate the timing of disease onset, recovery, or death during the epizootic. This allowed us to implement a framework based on epi-curves and Kaplan-Meier risk and survival analyses to assess the vulnerability of the coral community to SCTLD, and to provide a detailed prognosis of the outbreak's progression, magnitude and severity. Furthermore, by integrating the assessed epidemiological parameters into a multi-dimensional space, we outline a four-level framework that contributes to the characterization of species susceptibility to SCTLD. We believe our findings provide a novel perspective on understanding this highly aggressive disease, contributing to a more precise depiction of outbreak progression patterns, particularly in regions where only prevalence information (e.g., Fig 1) has been produced and, therefore likely aiding to understand the timing and implications of the outbreak more precisely.

## Methods

### Data collection

In order to describe the progression, magnitude and severity of the SCTLD epizootic, we conducted approximately biweekly evaluations of the coral communities at two sites afflicted by the SCTLD and located 1,100 m apart in the Puerto Morelos reef, Mexican Caribbean. One site, Bocana (20°52'29"N and 86°51'06"W) is located at the lagoon-ward end of an active channel. The other site, Picudas, (20°53'02"N and 86°50'54"W) experiences a more pronounced wave regime due to a lower reef crest and nearby a wide reef opening [26]. Both sites are 5m deep. Surveys started near the onset of the SCTLD epizootic and continued for one year (from June 2018 to July 2019), completing a total of 31 surveys across both sites.

At Bocana, four sampling points were established: B1 (453 $m^2$), B2 (19.5 $m^2$), B3 (140 $m^2$), and B4 (60 $m^2$). At Picudas, three points were defined: P1 (435 $m^2$), P2 (12 $m^2$) and P3 (9 $m^2$). These polygons were delimited by reef topography and visibility constraints, but remained fixed throughout the study. Despite differences in size, consistent effort was applied within each polygon to document all visible coral colonies across species during each visit through photographic surveys.

Initially, monitoring was focused on spatial disease patterns in *Pseudodiploria strigosa*, as part of a previous study on SCTLD dynamics [26]. However, during those surveys, photographic records were systematically taken of all visible coral colonies within the same sampling areas, regardless of species. Each colony was photographed individually during every

monitoring event, with multiple angles captured depending on its size and structural complexity. These images allowed us to track colonies over time and retrospectively assess their health status (healthy, diseased, or dead), visually estimate the percentage of dead tissue caused by SCTLD or previous mortality, and classify lesion progression type (acute, subacute, or chronic, sensu Work & Aeby [46]). Additionally, the maximum diameter of each colony at the onset of the study was measured using ImageJ (1.52a), based on scale markers included in the photographs.

As a result of these surveys, a total of 1,007 colonies of 26 coral species were recorded. Among these, 990 colonies belonged to 16 species that exhibited SCTLD signs in at least one individual during the study period (n = 422). Further analyses were conducted on these 16 species. Given that our previous approximation did not detect spatial differences in SCTLD distribution between sites [26], cases from both sites were merged to increase the sample size. This broader approach is further supported by Sharp et al. [25], who found that SCTLD transmission may not be associated with the density of susceptible individuals, reinforcing the rationale for merging data from both sites to improve statistical power.

**Individual disease status reconstruction**

To construct epidemic curves, determine daily status of prevalence and mortality, and to perform Kaplan-Meier risk and survival analyses, we needed to estimate when each affected colony began showing disease signs and when it either died or recovered. To achieve this, we reconstructed individual tissue loss trajectories using the percentage of dead tissue recorded for each affected colony (n = 422) at every monitoring event. The method used to estimate these trajectories depended on the number of available records per individual, resulting in four distinct methodological approaches: I) for individuals with more than two records; II) for individuals with exactly two records; and III) and IV) for individuals with only one recorded percentage of tissue loss. The details of each approach are described below.

I. For afflicted colonies that survived at least three observation periods (n = 123), we fitted individual Generalized Additive Mixed Models (GAMMs) for each colony, incorporating an autocorrelation structure (corARMA, corAR, or corCAR) selected based on the diagnostic residuals. To ensure model robustness, we assessed residual normality (Anderson-Darling test and histogram), homoscedasticity (White test), and the absence of autocorrelation (Durbin-Watson test). The results are provided in S3 Table. Given the limited number of observations, residuals were preprocessed using an iterative resampling approach (wild bootstrap). Finally, to determine the time of disease onset and complete tissue death, we interpolated the time points at which dead tissue reached 0.01% and 100%, respectively. This methodological framework accounts for the potential dependence between observations, as tissue loss percentages represent repeated measures within the same individual.

II. For diseased individuals recorded at only two time points (n = 81), we applied a deterministic method based on the linear equation:

$$y = m \cdot x + b \tag{1}$$

where (y) represents the percentage of tissue loss, (x) denotes time, (m) is the slope, and (b) is the intercept. The slope and intercept were calculated as follows:

$$m = \frac{y_2 - y_1}{x_2 - x_1} \tag{2}$$

$$b = y_1 - (m \cdot x_1) \tag{3}$$

The times corresponding to disease onset and complete tissue mortality were determined by solving for (x) in the linear equation, substituting the intercept and slope, and using (y) values of 0.01% for onset and 100% for mortality:

$$x = \frac{y - b}{m}$$

<div align="right">(4)</div>

For colonies with only one recorded observation of tissue loss, we acknowledge that direct estimation of lesion progression is not possible. To avoid excluding these cases, we implemented two approaches based on biologically informed assumptions and empirical reference data. These methods aim to reconstruct plausible trajectories while accounting for uncertainty, as detailed in points III and IV.

III.  The third approach was implemented for diseased colonies with only one recorded instance of tissue loss. These cases occurred either because the colonies were too small to relocate or because they died and were likely rapidly overgrown by algae, resulting in their capture as diseased during only one survey (n = 130). To address this limitation, we assumed that these individuals followed a similar trajectory to others of the same species, size category (small: < 33 cm; medium: > 33–66 cm; large: > 66 to <100 cm; and extra-large: > 100 cm), and lesion progression type. Based on this assumption, we first calculated individual tissue loss trajectories from colonies with sufficient temporal data, grouped by species, size category, and lesion progression type. From these trajectories, we derived an average daily tissue loss rate, which was then used to interpolate the percentage of tissue loss for colonies with only one recorded observation. This allowed us to estimate the full progression trajectory, including onset and extent of tissue loss, for single-observation cases. This procedure is illustrated in S1 Fig. A 95% confidence interval for each extrapolated trajectory was computed using the bootstrap percentile method and is provided for reference (S4 Table).

IV. For diseased colonies with only one tissue mortality record that could not be matched to any of the above categories of species, size, and lesion progression (n = 88), we estimated their individual tissue loss trajectories based on the colonization stage of the exposed skeleton. These stages (recent mortality, algal/microbial film, turf algae, and macroalgae) follow a consistent post-mortality succession (S2 Fig). Each stage was associated with an average transition time, calculated from a reference subsample of 157 colonies (S5 Table). Using this information, we inferred when each skeletal region was covered by living tissue. Colonies displaying multiple colonization stages provided several time points along the lesion progression. Depending on the number of inferred points, we then applied either a GAMM or a linear interpolation method (as in cases I and II, respectively) to reconstruct the full trajectory of tissue loss, from lesion onset to complete mortality.

All analyses were conducted in R using the following packages: *mgcv* v.1.9–3 [47] for GAMM modeling, *boot* v.1.3–31 [48] for bootstrap resampling, *car* v.3.1–3 [49] for White's test of heteroscedasticity, *nortest* v.1.0–4 [50] for the Anderson-Darling normality test, and *lmtest* v.0.9–40 [51] for the Durbin-Watson autocorrelation test.

## SCTLD epizootic progression and magnitude

Here two common epidemiological methods were applied to assess the vulnerability of coral community to SCTLD, and to provide a detailed prognosis of the epizootic's progression and magnitude: epidemic curves and Kaplan-Meier survival and risk analyses. These curves were computed both by individual species and by species grouped accordingly to their expected susceptibility (high, intermediate and low; *sensu* FDEP [30] and Papke et al., [33]). This analysis evaluates whether species grouped by expected susceptibility-based groups exhibit consistent differences not only in period prevalence and cumulative mortality, but also in other epidemiological indicators throughout the course of the epizootic. The list of species by group is provided in Table 1.

Species are grouped according to their expected susceptibility sensu [30] and [33]. AGRRA codes in brackets.

## Epizootic curves

Typically, the progression of an epizootic is characterized by successive stages: onset, increase in cases, peak, decline, and finally, stabilization or elimination of incidence [41]. This progression is depicted in the curve of incidence over time,

**Table 1. Epizootic parameters used to assess species susceptibility to SCTLD.**

| Expected suscepti-bility | Species | N | Time at first (days) | Time at 5% prevalence (days) | Epizootic peak time (days) | Outbreak length (days) | Population survival probability | Diseased survival probability | Risk at end | Period preva-lence (%) | Cumula-tive mor-tality (%) | Incidence rate at peak (%) |
|---|---|---|---|---|---|---|---|---|---|---|---|---|
| High | *Colpophyllia natans* (CNAT) | 17 | 52 | 52 | 92 | 181 | 0.07 | 0 | 0.92 | 100 | 100 | 27.3 |
| | *Dichocoenia stokesii* (DSTO) | 6 | 23 | 23 | 71 | 51 | 0 | 0 | 1 | 100 | 100 | 66.7 |
| | *Eusmilia fastigi-ata* (EFAS) | 1 | 24 | 24 | 24 | 1 | 1 | 1 | 1 | 100 | 0 | 100 |
| | *Meandrina jack-soni* (MJAC) | 2 | 27 | 27 | 29 | 10 | 0 | 0 | 1 | 100 | 100 | 50 |
| | *Pseudodiploria clivosa* (PCLI) | 5 | 1 | 1 | 1 | 114 | 0.6 | 0 | 0.4 | 60 | 60 | 20 |
| | *Pseudodiploria strigosa* (PSTR) | 198 | 3 | 39 | 99 | 390 | 0.05 | 0.006 | 0.87 | 95.5 | 94.9 | 7.6 |
| Interme-diate | *Montastraea cavernosa* (MCAV) | 56 | 56 | 78 | 169 | 228 | 0.34 | 0 | 0.56 | 66.1 | 66.1 | 13.3 |
| | *Orbicella annu-laris* (OANN) | 12 | 87 | 87 | 92 | 50 | 0.62 | 0.42 | 0.63 | 66.6 | 40 | 9.1 |
| | *Orbicella faveo-lata* (OFAV) | 36 | 55 | 71 | 113 | 276 | 0.62 | 0.26 | 0.44 | 50 | 36.1 | 9.4 |
| | *Stephanocoenia intersepta* (SINT) | 18 | 117 | 117 | 120 | 172 | 0.64 | 0 | 0.20 | 27.8 | 27.8 | 5.6 |
| | *Siderastrea siderea* (SSID) | 211 | 21 | 69 | 99 | 264 | 0.13 | 0.06 | 0.76 | 89.6 | 83.9 | 12.8 |
| Low | *Agaricia agaric-ites* (AAGA) | 255 | 12 | 109 | 113 | 322 | 0.70 | 0.08 | 0.21 | 27.5 | 25.1 | 2.1 |
| | *Agaricia tenuifo-lia* (ATEN) | 46 | 88 | 115 | 92 | 285 | 0.52 | 0.11 | 0.37 | 43.5 | 39.1 | 2.6 |
| | *Isophyllia sinu-osa* (ISIN) | 8 | 184 | 184 | 190 | 1 | 0.87 | 0 | 0.12 | 12.5 | 12.5 | 12.5 |
| | *Porites astreoi-des* (PAST) | 101 | 44 | 104 | 288 | 248 | 0.76 | 0.23 | 0.15 | 21.8 | 16.8 | 4.2 |
| | *Porites porites* (PPOR) | 18 | 285 | 285 | 288 | 1 | 0.83 | 0 | 0.07 | 11.1 | 11.1 | 7.1 |

known as an epizootic curve (epi-curve). Epi-curves by species and by susceptibility group were plotted using the *Inci-dence* R package [52]. These curves are generated using a list of the afflicted colonies by species and their corresponding date of signs onset. To highlight patterns in case emergence, epi-curves were constructed as weekly histograms. To esti-mate 95% confidence intervals (CI) for epizootic peak dates for each group, the original dates by species or groups were resampled with using the bootstrap method with replacement (1000 iterations; alpha = 0.05). The first highest mode in the bootstrap data was selected as the peak date.

After observing secondary waves in the combined epi-curve, we hypothesized that these might be linked to the disper-sal of the causative agent from sloughed tissue of dying colonies, which are potentially laden with microorganisms that could contribute to the feedback of the epizootic [30,33,53]. To explore whether mortality could help explain the observed incidence patterns, we visually compared the temporal alignment of daily incidence and mortality curves by overlaying their kernel density distributions. Additionally, to rule out the possibility that the pattern resulted from site aggregation, we compared incidence and mortality distributions by site.

## Risk and survival Kaplan-Meier curves

To assess the vulnerability of the coral community to SCTLD during the epizootic, Kaplan-Meier risk and survival curves were calculated for all susceptible species, as well as for each susceptibility group using the *survival* R package [54]. This aimed to determine the likelihood of exhibiting SCTLD signs during the epizootic (epizootic risk), the overall survival probability of the population, and the survival probability specifically among the affected population. Curves were estimated using the time in days to event occurrence (death or disease) and considering censored cases, i.e., cases in which the event has not occurred during study period, either because they remain healthy to the end of study, or they are not found from a certain moment. Survival probabilities were calculated by:

$$S_{(t)} = S_{(t-1)} * \left(1 - \frac{C}{n}\right)$$

(5)

where ($S_{(t-1)}$) is the survival probability at the immediately preceding time point, (C) is the number of cases (deaths) at the current time, and (n) is the number of individuals at risk at that time point. In parallel, the probability of becoming diseased (epidemic risk) was calculated as:

$$R = 1 - S_{(t)}$$

(6)

where (C) now represents new disease cases. In both cases, censored individuals are included in (n) up to the time of censoring, after which they are removed from the set of at-risk individuals. To test significant differences between groups, a 95% CI for each curve, and a Log-rank test were performed. If differences were found, a post hoc pairwise comparison was implemented with the Log-rank test, using the Benjamini-Hochberg method for adjusting p-values with the *survminer* R package [55].

## Prevalence and mortality

Furthermore, daily prevalence and cumulative mortality were calculated and plotted to track their changes throughout the epizootic, and to compare these trends with the epidemic curves. Prevalence was calculated daily as:

$$P = \frac{d}{n + d} * 100$$

(7)

where (d) is the number of diseased individuals on the day, and (n) is the total number individuals at risk on the same day. Cumulative mortality was calculated daily as:

$$M = \frac{D}{T} * 100$$

(8)

where (D) is the total number of cumulated individuals dead by SCTLD, and (T) is the total number of individuals (deaths included).

## Species susceptibility to SCTLD

In coral diseases ecology, susceptibility is commonly defined as the likelihood of corals becoming diseased or dying due to specific stressor, and prevalence is a common (and perhaps the most frequent) measure to indicate it at the population and community levels [56,57]. However, prevalence reflects only the proportion of diseased individuals at a given time, and it does not capture the temporal variability of disease risk. While often interpreted as the probability of becoming ill (i.e., epizootic risk), prevalence more accurately represents the probability of finding a diseased individual within the population. Therefore, to better characterize species vulnerability to SCTLD during the epizootic, we employed ten indicators,

derived from the epi-curves, prevalence, mortality, and survival analysis. These metrics encompass the onset, progression, and cumulative impact of the outbreak for each coral species.

## Susceptibility at the beginning of the epizootic

The start of the SCTLD outbreak in our study was defined as the date of the first diseased individual observed. To determine the sequence of appearance of SCTLD between species, the time elapsed between the start of the SCTLD outbreak, and the first diseased individual of each species was calculated (*time at first*). This metric has been used to infer relative susceptibility, assuming that species showing signs earlier are more vulnerable [23,30].

In parallel, we estimated the time at which each species reached a prevalence of 5%, which we used as a population-level reference to compare detection dynamics. This value is informed by historical patterns in the Caribbean, where coral diseases typically remain below 5% under endemic conditions [58]. We hypothesize that traditional sampling methods tend to detect disease only after it exceeds such visibility thresholds, rather than capturing the earliest affected colonies.

## Susceptibility during the progression of epizootic

To assess species-specific susceptibility throughout the epizootic, we calculated three metrics:

-Outbreak length: Time between the first and last observed case per species.

-Time to peak: Days from outbreak onset to the species-specific peak.

-Height of peak: Proportion of new cases at peak time relative to the population at risk, calculated as:

$$I = \frac{Ip}{n + Ip} * 100$$

$$(9)$$

where (Ip) is the number of new cases at the peak, and (n) is the number of individuals at risk at that time.

## Cumulative impact after epizootic

To evaluate the overall impact and severity across coral populations, we calculated:

-Epizootic risk at end (Kaplan-Meier estimates).

-Population and Diseased survival probabilities at end (Kaplan-Meier estimates).

-Period prevalence (*Pp*), adjusted for censored individuals using species-specific risk estimates:

$$Pp = \frac{Td + (c * R)}{N} * 100$$

$$(10)$$

-Total mortality (*M*), incorporating expected deaths among censored individuals:

$$M = \frac{TD + (d * S)}{N} * 100$$

$$(11)$$

where (Td) and (TD) are the observed diseased and dead individuals, (c) and (d) are the censored individuals and expected diseased among them, (R) is the estimated risk, (S) is the probability of death among diseased individuals, and (N) is the initial sample size.

To test for species-level differences, we applied Kruskal-Wallis tests followed by Dunn post hoc comparisons. Species associations across metrics were visualized via Principal Component Analysis (PCA), performed using the *factoextra*

R package. All variables were centered (subtracting the mean) and scaled to unit variance (dividing by the standard deviation) prior to analysis to ensure equal contribution to the principal components regardless of their original units or ranges. *Eusmilia fastigiata* was excluded due to its low sample size (N = 1).

## Results

### Epizootic progression

The progression of SCTLD epizootic was analyzed using epi-curves (Fig 2) based on 422 coral colonies affected across 16 susceptible species. In addition, epi-curves were produced for all species together and separately in three groups of species based on their expected levels of susceptibility: high, intermediate and low (Fig 3; as noted by [30] and [33]). Species in each susceptibility group are shown in Table 1.

The progression of the epizootic began with a colony of *Pseudodiploria clivosa* (N = 5), estimated as the first diseased individual. The onset time and duration of the outbreak varied among coral species. Intermediate and low susceptibility species, such as *Siderastrea siderea* (N = 211)*, Agaricia agaricites* (N = 255) and *Porites astreoides* (N = 101), developed signs earlier than some high susceptibility species like *Colpophyllia natans* (N = 17). In *C. natans, P. strigosa* (N = 198)*, Orbicella faveolata* (N = 36)*, S. siderea* and *A. agaricites* the epi-curves exhibited a positive skew, with the central location to the left and the tail extending to the right, indicating a rapid initial spread. *Dichocoenia stokesii* (N = 6)*, Meandrina jacksoni* (N = 2), *P. clivosa* and *Orbicella annularis* (N = 12) also had a positive skew but with a limited number of cases. Conversely, *P. astreoides,* showed a negative skew, suggesting a slower increase of cases. *Agaricia tenuifolia* (N = 46)*, Montastraea cavernosa* (N = 56), and *Stephanocoenia intersepta* (N = 18) displayed a more dispersed distribution of cases over time. *Isophyllia sinuosa* (N = 8) and *Porites porites* (N = 18) each had only one case, occurring around the mid- and late-epizootic, respectively. For all species, except *P. astreoides,* peak occurrences were within the first few weeks of their epi-curves, but with varying times from the beginning of the epizootic (Table 1). Lastly, species with right-skewed epi-curves experienced a prolonged decline in cases, characterized by multiple waves (Fig 2). It is important to note that outbreak duration and curve shape may be influenced by sample size, particularly in species with limited representation.

When analyzing the curves by susceptibility group, all epidemic curves exhibited positive skewness with a prolonged tail to the right (Fig 3B-D). Cases in each susceptibility group began as early as the first month of the outbreak. A Kruskal-Wallis (K-W) test indicated that the timing of the onset of the first diseased colony was independent of the species group (K-W test, p = 0.08). In contrast, the species sequence based on 5% prevalence time showed significant differences between species groups (K-W test, p = 0.003). However, a pairwise post hoc analysis revealed differences only between the high-susceptibility and low-susceptibility groups (Dunn test, p = 0.002). Similarly, the time to epizootic peak was significantly different only between the high and low susceptibility groups (post hoc Dunn test, p = 0.02), with a difference of only over a month between peaks. No association between susceptibility groups and outbreak duration was detected (K-W test, p = 0.91), a result further explored in the discussion.

Combining data from all species reveals a curve slightly skewed to the left, with the peak observed at 99 days from the onset (Fig 3A). The presence of multiple waves following the peak, along with an approximately similar distribution of cases across all susceptibility groups, is particularly notable. When comparing the density distributions of daily incidence and mortality, we found that, after the epizootic peak, increases in mortality precede secondary waves, both in the combined data and in the separate data from each site (Fig 4).

### Vulnerability of the coral community

The vulnerability of the coral community to SCTLD was assessed using Kaplan-Meier risk and survival curves, calculated for all susceptible species (S3 and S4 Figs, respectively) and for each susceptibility group (Fig 5A-C). The results revealed that species generally exhibited vulnerability consistent with their susceptibility groups. Species within the high susceptibility group were the most vulnerable, reaching higher risk levels more rapidly and exhibiting near-zero survival

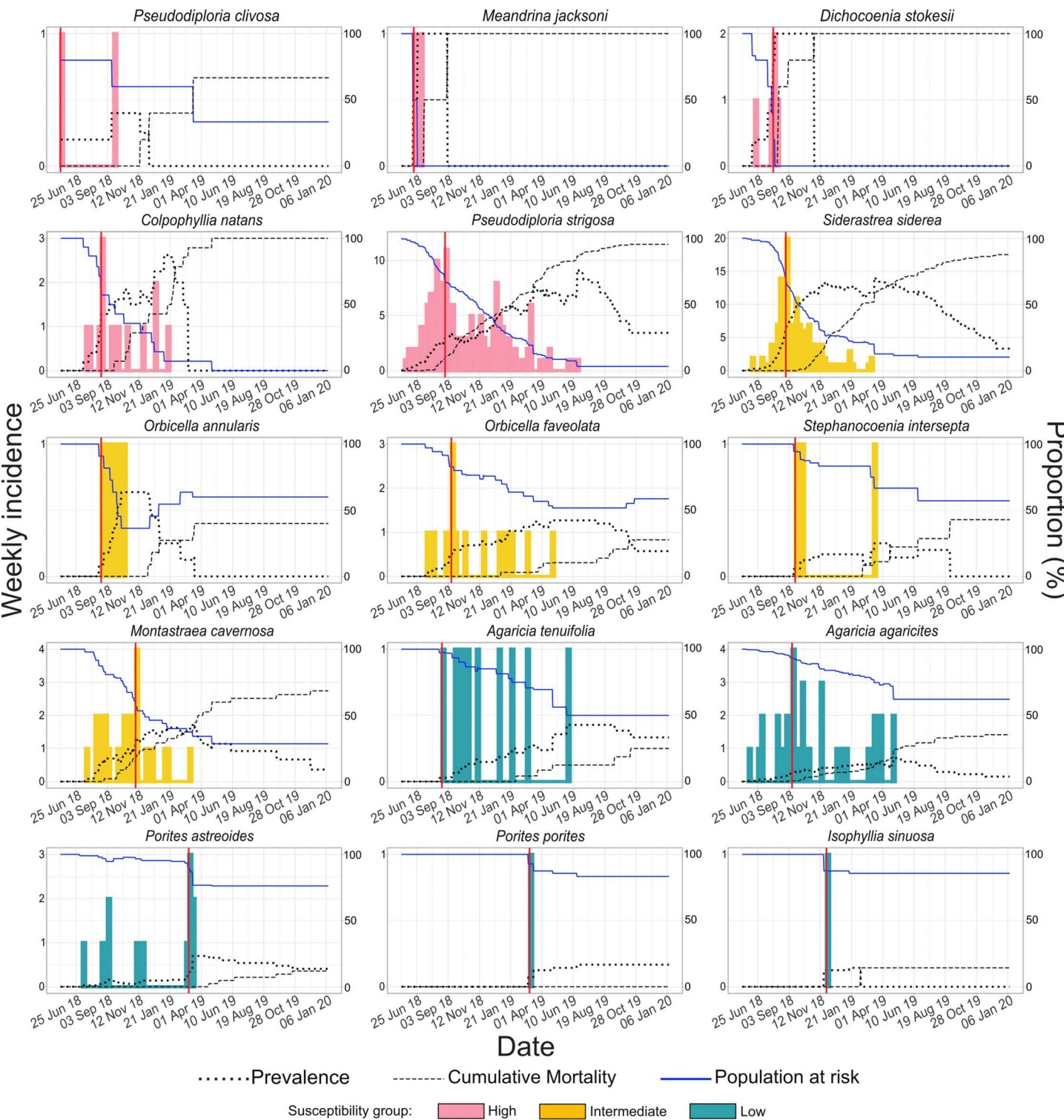

**Fig 2. Epi-curves and daily prevalence curves by species through the epizootic.** Incidence is presented weekly in colored bars (showing their expected susceptibility) with values on the primary axis. Sample sizes (N) per species are reported in Table 1. Red vertical lines mark the time of the epizootic peak for each species. Dotted black lines show the daily prevalence curve. Dashed black lines show the cumulative mortality percentage. Blue continuous lines show the proportion of the population at risk. These three proportions have scale on the secondary axis. Climbs on population at risk curves are due to recoveries while decreases after epi-curves' end are due to adjustments on population at risk proportion by censoring.

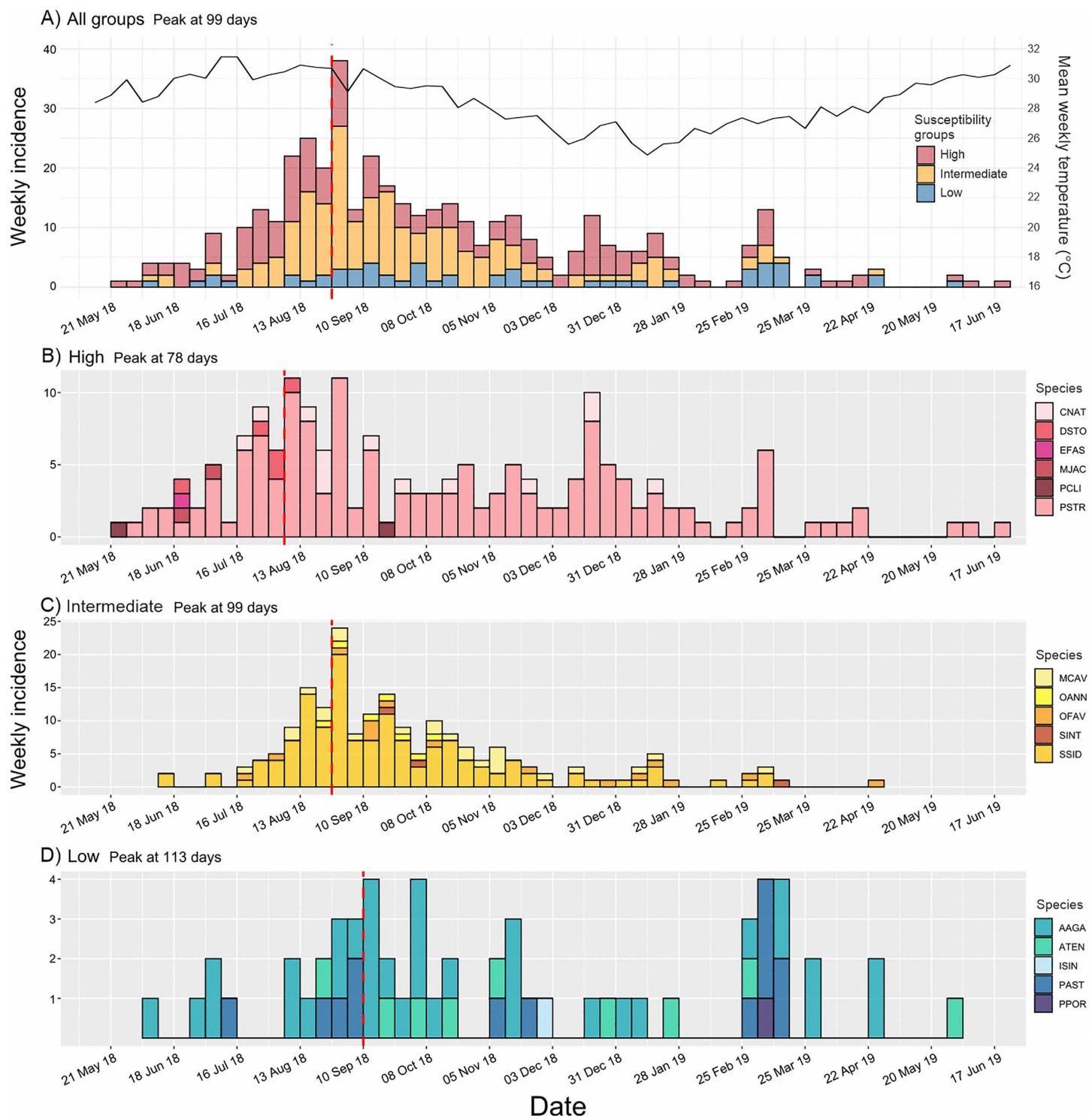

**Fig 3. Epi-curves grouping species.** A) All susceptible species contrasted by susceptibility groups. All groups started with signs within the first month. Mean weekly temperature (°C) is marked by a solid black line (scale on the secondary axe). B-D) Epi-curves by susceptibility group: B) high, C) intermediate, and D) low susceptibility. Weekly incidence and time in days are in the Y and X axis, respectively. Species in AGRRA code. Red dashed lines mark the epizootic peak.

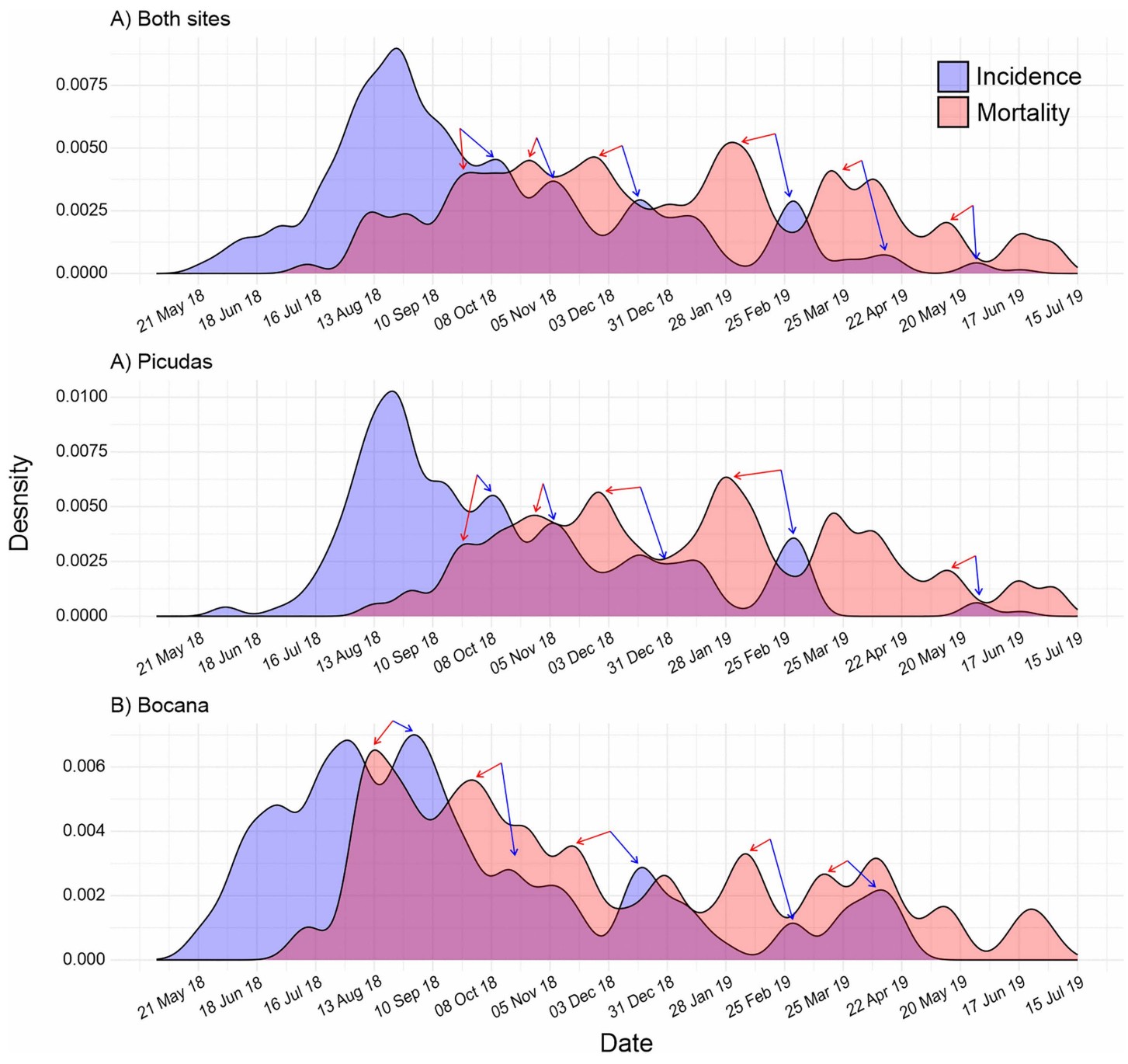

**Fig 4. Comparison of the density distributions of daily incidence and mortality during the SCTLD epizootic.** A) Both sites; B) Picudas site; and C) Bocana site. Every site presents around five minor waves of incidence (blue) preceded each by a peak of mortality (red), each signaled by a pair of arrows in blue and red, respectively.

probabilities for both at-risk and diseased populations, except for *P. clivosa* which never reached the 50% risk probability. Interestingly *S. siderea,* despite belonging to the intermediate susceptibility group, showed a 76% risk probability and very low survival probabilities (under 15%) by the end of the epizootic. Despite showing moderate to high survival probabilities

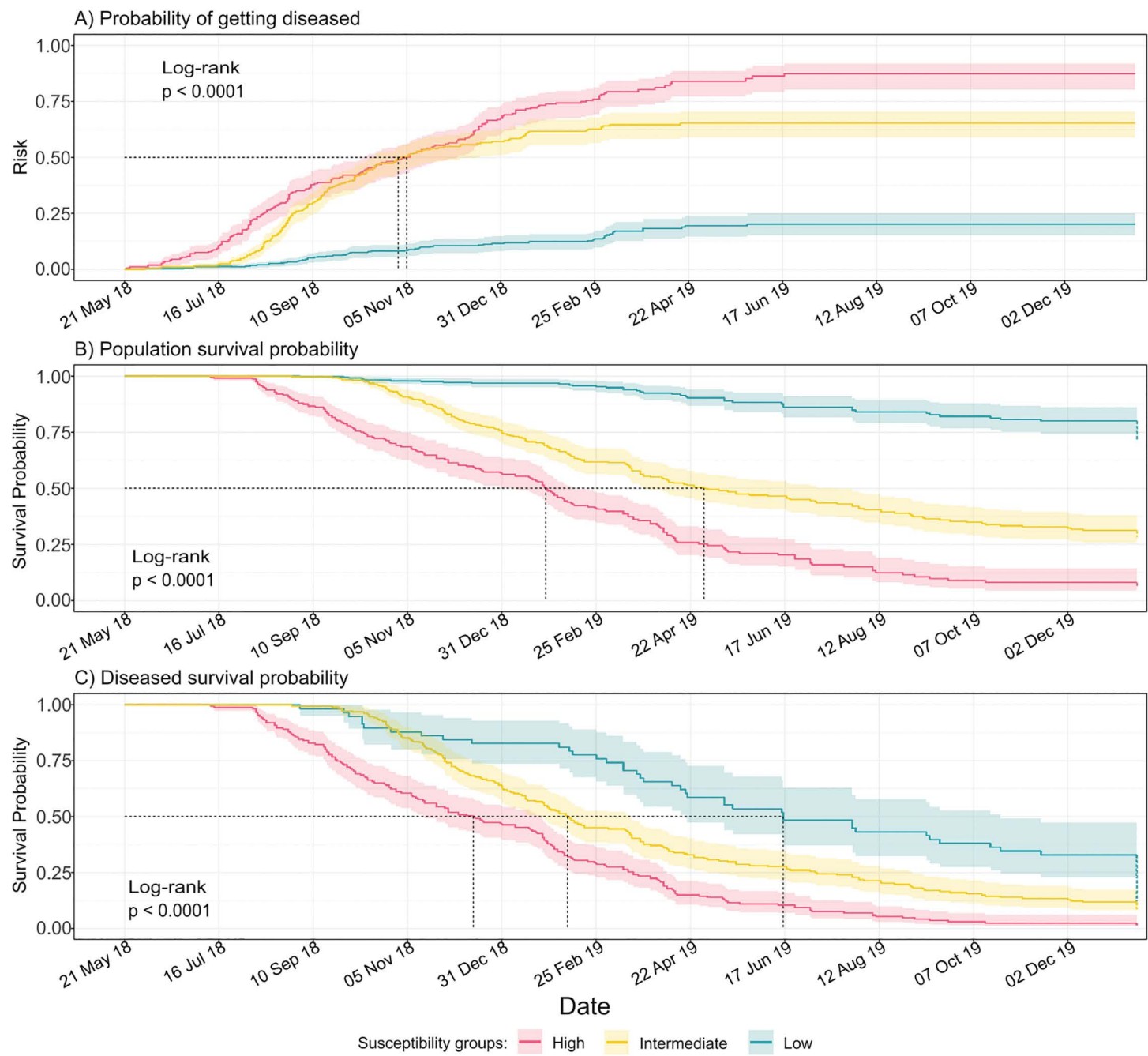

**Fig 5. Kaplan-Meyer Risk and Survival curves for the susceptibility species groups.** A) Risk curve showing the probability of disease over time. B) Survival probabilities for all individuals and C) Survival only of those with SCTLD. Dashed lines mark when 50% experienced the event. Shaded ribbons represent 95% confidence intervals. P-values from log-rank tests: χ² = 263, 297, and 63.1, respectively. Data is shown up to day 600 since outbreak onset. In B and C, decreases in survival probabilities occurred beyond day 600 are depicted as vertically dashed lines. These lines do not indicate sudden drops at that point but summarize posterior mortality up to 2.8 years.

in their at-risk populations (S4 Fig), the species *A. agaricites*, *A. tenuifolia*, *I. sinuosa*, *S. intersepta*, and *P. porites* exhibited notably low survival (below 11%) in their diseased populations.

Despite these exceptions, comparison of the three risk curves showed significant differences between all species groups (Fig 5A; Log-rank test, $p < 0.0001$), further supported by post hoc pairwise comparisons ($p < 0.001$). Despite these differences, there was an overlap in the risk curves for high and intermediate susceptibility groups between September and December 2018, both reaching 50% risk around the first week of November 2018. Contrastingly, the low susceptibility group never reached a 25% risk.

Survival probability curves differed significantly among susceptibility groups for both the populations at risk and the diseased populations (Log rang test: $p < 0.0001$; pairwise post hoc: $p < 0.001$), with lower survival in the high susceptibility. However, differences among diseased populations were smaller ($\chi^2 = 62.5$ vs. 296; Fig 5C), and by the end of the follow-up, when all unrecovered colonies had died, no significant differences remained (K–W test, $p = 0.22$).

## Magnitude & Severity of epizootic

The impact of the SCTLD epizootic on coral populations was assessed by period prevalence, cumulative mortality and peak incidence rate (height of peak; Table 1). Although daily prevalence curves increased toward the end of the outbreak in all species, this rise is partly driven by the declining number of at-risk colonies (Fig. 2), including censored individuals, which inflates daily prevalence values. Therefore, period prevalence was used as a more reliable measure of epizootic magnitude during the final phase.

The high-susceptibility species were the most affected, with the higher peak heights, period prevalence over 93%, and cumulative mortality above 91%. Except *P. clivosa*, which had 60% in both metrics. Interestingly, *S. siderea*, an intermediate susceptibility species, exhibited high period prevalence and mortality (90% and 84%, respectively) (Table 1).

All magnitude and severity metrics showed significant differences only between high and low susceptibility groups (pairwise post hoc Dunn test, $p < 0.05$).

## SCTLD species susceptibility

As previously stated, no consistent differences were observed in the progression of the epizootic among affected species. However, differences in species susceptibility to SCTLD became evident when considering risk, magnitude, and severity. When species were grouped based on their expected susceptibility, more distinct patterns emerged (Fig 5). In contrast, ungrouped analyses revealed greater variability in species responses (S3 and S4 Figs; Table 1), particularly within the intermediate-susceptibility group.

To achieve a more comprehensive understanding of species susceptibility, we integrated the assessed epidemiological parameters (Table 1) using a principal component analysis (PCA, Fig 6). This multivariate approach enabled us to position species along a susceptibility gradient, divided into four vertical segments (Levels I–IV) from right (highest susceptibility) to left (lowest). We based this classification on the horizontal axis (PC1), which explains 60.9% of the total variance and captures the widest distribution of susceptibility-related indicators (e.g., risk, survival, mortality, prevalence). In contrast, the vertical axis (PC2) is primarily influenced by outbreak duration, a variable that may be confounded by species abundance. Therefore, we avoided quadrant-based classification, which could misrepresent susceptibility in species with longer outbreaks but higher population sizes. Overall, the distinction between highly and low susceptible species was largely consistent across the gradient. Nevertheless, the species *S. siderea* (intermediate expected susceptibility), *A. tenuifolia* and *A. agaricites* (both low expected susceptibility) demonstrated a tendency towards increased susceptibility, highlighting the value of integrating multiple indicators.

Below, we summarize the characteristics of each PCA-derived susceptibility level:

**Level I:** (*D. stokesii, M. jacksoni, C. natans* and *P. strigosa*): Early onset and rapid progression to epizootic levels. High total prevalence and mortality. Risk near 100%, survival near 0%. Outbreak length varies with population size; shorter

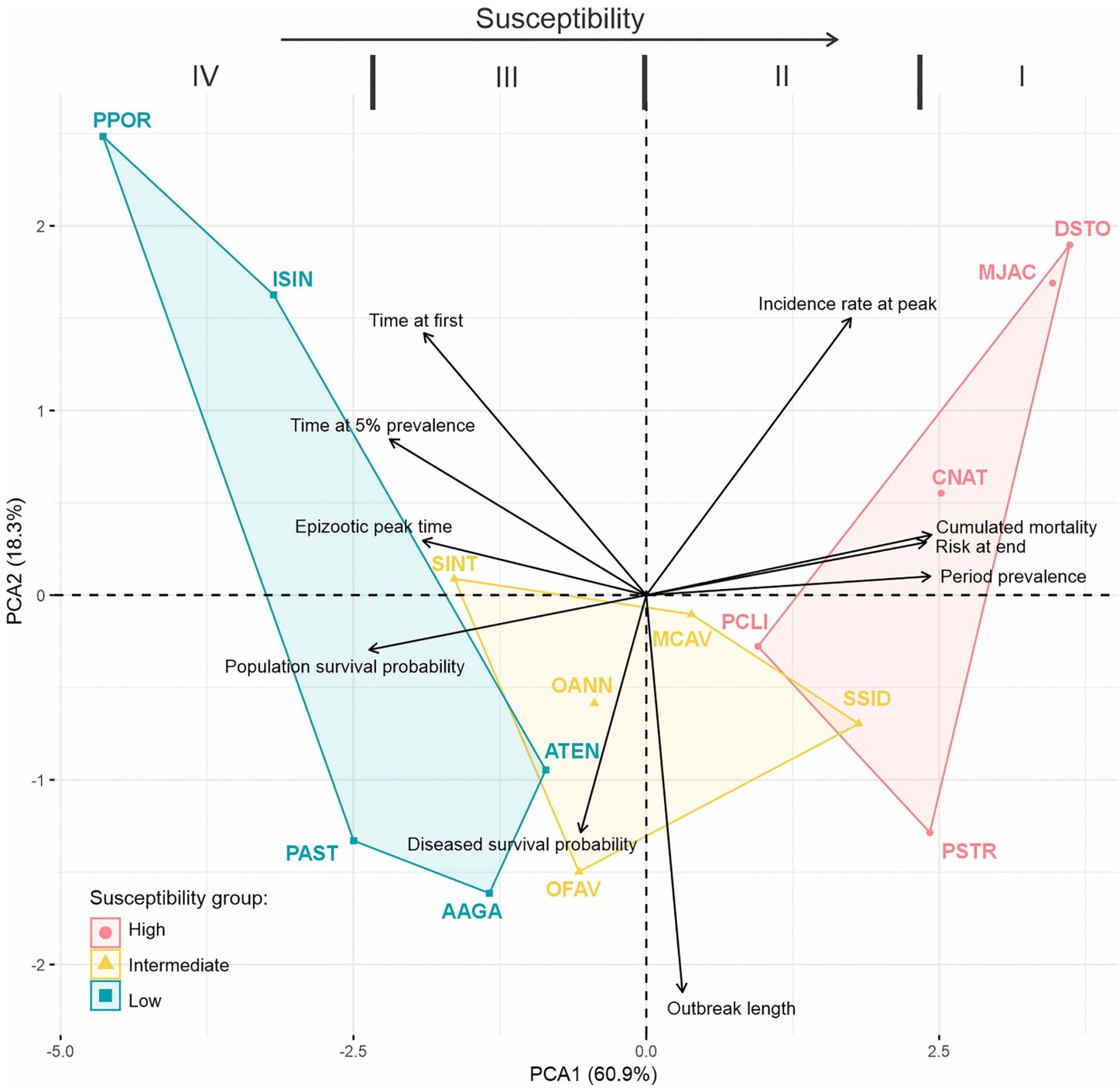

**Fig 6. Principal component analysis (PCA) of ten SCTLD epizootic indicators across coral species.** Species are grouped into colored polygons based on their expected susceptibility: red (High), yellow (Intermediate), and blue (Low). Roman numerals indicate exploratory susceptibility levels derived from the visual distribution of species in PCA space. Higher susceptibility is positioned towards the right along PC1. Species names in AGRRA code.

outbreaks show higher peaks and faster spread, while longer outbreaks exhibit lower peaks and multiple transmission waves.

**Level II:** (*S. siderea, P. clivosa* and *M. cavernosa*): Early to intermediate onset. Moderate to high total prevalence, mortality and risk. Low survival of the diseased population but moderated survival of the general population. Early peak time with intermediate peak height. Initial rapid spread followed by prolonged transmission.

**Level III:** (*O. annularis, O. faveolata, S. intersepta, A. tenuifolia,* and *A. agaricites*): Variable onset progression. Moderated total prevalence and mortality. Intermediate to low risk. Moderate to high population survival but reduced survival of diseased individuals (except for *O. annularis, O. faveolata*). Variable outbreak length related to population size; shorter outbreaks show higher peaks; longer outbreaks show slower spread.

**Level IV:** (*P. astreoides*, *P. porites* and *I. sinuosa*): Late onset and low mortality and total prevalence. Very low risk and high population survival. Variable outbreak length related to the population size. Short peak height in all cases.

## Discussion

In this study, we integrated epidemiological tools widely applied in other systems but largely overlooked in coral epizootic research, to increase our understanding of the temporal dynamics of the SCTLD epizootic. This approach enabled us to determine that species-specific susceptibility was associated with the risk, magnitude, and severity of the epizootic, but not with its progression. We outline a four-level framework of coral susceptibility to SCTLD based on diverse epidemic indicators, rather than solely on prevalence measures. This framework revealed a higher-than-expected susceptibility in *S. siderea*, *A. agaricites* and *A. tenuifolia*. Moreover, by examining the detailed progression of the epizootic, we identified that secondary waves were preceded by increases in mortality. These findings highlight the importance of comprehensive epidemiological studies in investigating coral disease dynamics. Overall, our results contribute to assessing coral community vulnerability during epizootics, interpreting epidemic curves, and exploring their implications for the etiology of this coral syndrome.

### Evaluating the vulnerability of coral community during epizootics

One of the most significant findings of this study is that species susceptibility to SCTLD was reflected in the epidemic´s risk, magnitude and severity, but not in its temporal progression. This became more evident when comparing epi-curves across susceptibility groups. In this regard, previous studies have often interpreted the sequence of disease onset as a proxy for susceptibility [33,59]; however, our data show that several intermediate- and low- susceptibility species exhibited signs earlier than some highly susceptible ones.

Interestingly, when analyzing the time to reach 5% prevalence, our results aligned more closely with species-specific onset patterns reported in earlier work [30]. This may reflect variations in sample size, species composition at each site, and the spatial distribution of cases, as traditional sampling methods have demonstrated low accuracy in detecting coral diseases [60]. In contrast, our study employed a comprehensive sampling approach, covering approximately 1100 m$^2$ per site, whereas SCTLD studies typically cover areas of 20–200 m$^2$ [7,23,25,61,62]. Thus, intermediate and low susceptibility species might not have been detected until their populations reached epizootic levels.

Another key aspect emphasized in this study is the limited attention given to the progression of coral epizootics. Most studies on coral diseases assess population or community vulnerability using prevalence-based metrics [32,35,44,63]. However, our results indicate that prevalence may not be a reliable measure of species susceptibility during epizootics (Fig 1). To explore this further, we sub-sampled daily prevalence at days 100, 200, 300, 400, 500, and 600, comparing values among species (S5 Fig). We found that, even if the epizootic started at roughly the same time across populations, prevalence comparisons differed at each time point. This effect would likely be more pronounced between populations from different places.

Therefore, determining coral susceptibility should involve more than assessing population-level impact. When integrating multiple epizootic measures, such as progression, risk, magnitude, and severity, some differences are observed with

respect to the expected patterns (Fig 6). Notably, *S. siderea*, *A. agaricites*, and *A. tenuifolia* showed greater vulnerability than previous reported [8,29]. These species, particularly *S. siderea*, have exhibited variability in prevalence reports (e.g., [24,63]), which may reflect environmental variation, regional stress histories [33], the presence of multiple lineages [64–66], co-occurrence of multiple pathologies [67], or the application of a more comprehensive methodology in this study.

Another important aspect to consider is that in all high-susceptibility species analyzed in this study (except *P. clivosa*), the cessation of the outbreak may be associated with the depletion of the at-risk population (blue lines in Fig 2). In contrast, for species in intermediate- and low-susceptibility groups, where the termination of the outbreak was not due to the depletion of the population at risk but rather to the cessation of transmission. Therefore, we propose that outbreak duration is not a reliable measure of susceptibility in a coral population with low abundance. The cessation of transmission in intermediate- and low-susceptibility species could be explained by the saturation of local susceptible individuals [68]. That is, as corals die, the density of susceptible individuals decreases, limiting further transmission [69]. However, previous studies have observed the development of SCTLD independently of coral [25,26]. Therefore, it appears that unaffected colonies demonstrated greater resistance to SCTLD.

### Insights from epizootic curves into the etiology of SCTLD

The shape of the epi-curve with all species combined (Fig 3A and Fig 4A), suggests that the SCTLD epizootic may be linked to exposure from a common point source with potential secondary transmission events [41,43]. Assuming rapid transmission facilitated by currents [53], simultaneous exposure could explain the temporal alignment of onset and peak across susceptibility groups. Williams et al. [23] reported a similar alignment of epizootic peaks across different reef zones in the Lower Florida Keys, despite differences in onset timing, suggesting that similar patterns may occur at other SCTLD-affected sites.

Interestingly, secondary transmission events appear related to mortality, as we have found that increases in mortality preceded secondary waves of the epizootic after the epizootic peak. This pattern was consistent across sites (Fig 4B, C). Previously, it has been raised that the dispersal of SCTLD could be favored by the sloughed tissue coming from dying colonies, as it is potentially loaded with microorganisms [15,33,53]. While our findings do not constitute direct field evidence, they provide model-based support for this hypothesis. Further investigation is needed to establish causality, but the observed patterns are consistent with a potential feedback loop in which coral tissue mortality contributes to increase transmission, thereby aggravating and prolonging the epizootic. This feedback loop has the potential to serve as a transmission mechanism and could even shed light on the origin of the epizootic.

This result could imply both the existence of a primary pathogen and opportunistic infections driven by the large amount of tissue shed from dead corals. Several bacterial types and viral infections of symbiotic algae have been proposed as potential etiological agents of the SCTLD (e.g., [13–17,70–72]). However, their diversity, variability across sites and host [18,21,33], combined with their absence of direct involvement in tissue damage as evidenced by histopathological analyses [19–20], indicate they may act opportunistically [20,21,71]. Such microbial complexity could explain the variability in histopathological lesions observed across outbreaks [19,20,73], potentially reflecting differences in microbial exposure, dysbiosis, or secondary infections that influence the clinical presentation of SCTLD.

On the other hand, in the person-to-person mode of transmission in human epidemics, intervals between epi-curve crests often correspond to incubation periods [41]. In our curves, intervals between waves were approximately 6–13 weeks. But, assuming the role of mortality in producing secondary waves, the lag between mortality peaks and subsequent wave crests was 2–4 weeks. However, in experimental assays, incubation periods ranging from 4–20 days [27,74]. These differences could be due to the complex interactions between hosts (holobiont), pathogens and their environment at the reef scale, which are difficult to reproduce experimentally. This suggests that SCTLD transmission may not follow colony to colony dispersal but rather exposure from a common source, consistent with findings that SCTLD epizootics may not rely on colony density [25,26].

Additionally, while not mutually exclusive, this phenomenon may be linked to variations in environmental conditions, such as annual temperature patterns or thermal anomalies. Our findings suggest a positive correlation between the increase in cases and rising temperatures, alongside a decline in cases during the main wave with the onset of winter (Fig 3A). Conversely, secondary waves began to emerge starting in the winter months. Further research is required to investigate the potential relationship between our data and thermal anomalies. Notably, in other regions, the association between SCTLD and temperature has been inconclusive, with both positive [7,31] and negative [10,23] correlations reported in relation to the epizootic's progression. These findings suggest that the susceptibility of coral to disease may be influenced by a range of environmental factors, not limited to elevated temperatures. In the other hand

However, if all coral colonies were exposed to SCTLD agents simultaneously, as suggested by our observations, the variation in disease risk among species implies that susceptibility is modulated by intrinsic biological or ecological factors. This variability in the susceptibility of corals has indeed been observed within populations of the same species [26,75], and inclusively between clonal individuals [76]. Resistance or susceptibility could be mediated by the size and morphology of coral colonies [75]. More recently, it has been suggested that the type of symbiont (genus) dominant in a coral species or individuals may be very closely related to susceptibility to SCTLD. For instance, intraspecific susceptibility in *O. faveolata* has been found to be more closely linked to the genus of the symbiont rather than the genotype of the host [76]. Furthermore, a hierarchy of susceptibility to SCTLD has been observed among symbiont genera [77]. Histopathologic analyses have observed that the first tissue layer affected is the gastrodermis, where algal symbionts reside [19,20]. Beavers et al. [78] observed that SCTLD-affected colonies phagocytize dysfunctional symbionts. They propose that viral infections of symbiont algae trigger immune responses against affected symbionts or their pathogens, where the symbiont genera show a differential resistance to infection and degradation. However, more research is needed to state the role of viruses as primary causative agents or opportunistic pathogens [21,33].

Opportunistic infections may contribute to the onset of the epizootic through diverse, non-exclusive pathways. A substantial influx of microorganisms, pathogenic or not, can induce tissue necrosis by colonizing lesions or being ingested [79]. Additionally, they may alter and disrupt microbial assemblages in the coral's mucus layer [20], leading to dysbiosis and the development of opportunistic infections within the microbiome [80–82]. These conditions may trigger immune responses such as peroxidase activity, melanization, and apoptosis, as documented in SCTLD [33,83]. Which, when dysregulated, exacerbate tissue damage and destabilize coral-algal symbiosis [84]. Corals hosting stress-tolerant symbionts may be more likely to recover due to preserved nutritional resources.

An outbreak caused by opportunistic microorganisms traveling in the organic matter detached from dead corals, does not contradict the possibility of large-scale contagious dispersal favored by the marine currents [22,53]. Further research is necessary to elucidate the dispersal patterns of SCTLD at a regional scale within the Mexican Caribbean. Additionally, potential sources of dispersion such as transport from Florida reefs [22], discharge from ballast water [24,85] and resuspended sediments [74] should be carefully considered.

## Methodological considerations and limitations

This study presents several methodological considerations: Variation in sample sizes across species, may influence the shape and duration of epidemic curves, particularly in underrepresented taxa. However, near-complete censuses within defined survey areas, recording all visible coral colonies regardless of species or condition, minimized sampling bias and reduced the likelihood of abundance-driven artefacts. Notably, *M. jacksoni* (N=2) and *D. stokesii* (N=6) showed extreme susceptibility (100% prevalence, mortality, and risk), consistent with reports from other regions [30,33], while other low-abundance species such as *P. porites* (N=18) and *I. sinuosa* (N=8) exhibited much lower susceptibility (<13%). These patterns suggest that species-level vulnerability reflects intrinsic biological differences rather than sample size or visibility; For colonies with only one recorded observation of tissue loss, inference-based approaches (III and IV, see methods) were applied transparently, incorporating ecological indicators and uncertainty. Among these single-record

cases, the distribution across susceptibility groups was approximately balanced (57% high, 44.5% intermediate, 62% low; relative to diseased colonies in each group), suggesting no systematic bias toward any group. A more influential factor was colony size: 61% of these cases corresponded to small colonies. We consider that this approach adds value by recovering information that would otherwise be excluded and by enabling the inclusion of these cases in broader epidemiological analyses; To improve temporal resolution, data from two nearby sites were pooled under the assumption of non-density-dependent dynamics, enhancing curve continuity while potentially obscuring fine-scale spatial patterns; Finally, extrapolation of tissue loss timing via algal colonization did not account for skeletal morphology, though environmental conditions were assumed to dominate secondary succession.

## Perspectives and implications

This study highlights the value of epidemic curves as a central tool for characterizing coral disease dynamics. Their ability to capture the temporal structure of outbreaks makes them especially useful for detecting shifts in disease behavior and identifying outbreaks that may escalate into epizootics. Given their ease of implementation and interpretability, epi-curves can be integrated into routine monitoring programs to provide real-time insights and support timely decisions on surveillance, containment, and restoration. When combined with risk-based indicators, they offer a robust framework for adaptive management, helping practitioners anticipate critical transitions and prioritize interventions. Although applied here during an active outbreak, this approach can be extended to long-term monitoring, generating high-resolution temporal data that reveals subtle changes in disease dynamics and enable links to environmental or anthropogenic drivers. Moreover, the use of epidemic curves and integrated indicators is scalable to other coral diseases and marine taxa affected by epizootic events. As SCTLD continues to expand across the Caribbean, these tools could enhance regional coordination, optimize resource allocation, and strengthen conservation responses.

## Conclusions

This study demonstrates the value and utility of epizootic curves for investigating coral disease epidemics, offering a significant advantage over other commonly applied tools, such as prevalence measurements. By integrating this approach with assessments of the risk, magnitude, and severity of the epizootic, we were able to explore relevant aspects of its progression, generate insights into the potential etiology of SCTLD, and examine patterns of species-specific susceptibility. Notably, species-specific susceptibility was found to influence the risk, magnitude, and severity of the epizootic, although it did not affect its progression. This comprehensive, fine-grained methodology also revealed greater susceptibility in certain species that had previously been underestimated. Finally, this study provides evidence of a possible feedback loop acting in the SCTLD epizootic, which has never been described before. Based on these results, we strongly recommend integrating diverse epidemiological evidence, particularly epidemic curves, into current monitoring efforts. Additionally, we suggest investigating whether an epizootic progression pattern similar to the one observed here has occurred at other sites affected by SCTLD, and other coral diseases or syndromes.

## Supporting information

**S1 Fig. Estimation of lesion progression for colonies with only one recorded tissue loss value.** Example for *P. strigosa* (medium size, acute lesion progression, n = 12): A) Daily tissue loss trajectories from colonies with multiple observations were grouped by species, size category, and lesion progression type. Each colored line is an individual trajectory. B) Time was normalized around the 50% tissue loss point to align trajectories and estimate average trajectory. C) The average trajectory (blue line) and its 95% confidence interval (blue ribbon) were computed via bootstrap percentile method. D) Red dots show single-record colonies; their lesion progression was extrapolated using the reference trajectory, assuming similar dynamics.
(PDF)

**S2 Fig. Identification of the stages of colonization of the denuded coral skeleton by SCTLD.** Process illustrated with two colonies of *P. strigosa*. Dual view per colony: natural appearance (top) and colonization stage overlays (bottom). Legend at the lower right corner. Colonization stages are arranged outward from the living tissue and delineated by distinct line patterns in the following order: Recently dead (almost pure white coloration), Film 1 (incipient microbial film), Film 2 (well-established microbial biofilm; denser algal films with more saturated hues), Film 3 (incipient algal mat; filamentous algae beginning to establish, with possible sediment entrapment), Turf algae (denser mats of juvenile macroalgae or filamentous algae), and Macroalgae (advanced macroalgal growth, often identifiable to genus level).
(PDF)

**S3 Fig. Probability of becoming diseased by species.** Colored bands represent the 95% Confidence Interval. Colors indicate the expected susceptibility groups: red for high susceptibility, yellow for medium susceptibility, and blue for low susceptibility. A notable decrease in risk is observed as one moves to lower levels of expected susceptibility. However, some species, such as *P. clivosa, S. siderea*, and *S. intersepta*, display values that deviate from their expected susceptibility groups. Overall, the medium susceptibility group exhibited greater heterogeneity.
(PDF)

**S4 Fig. Kaplan-Meier survival probability curves for the population (blue lines) and diseased individuals (red lines) by species.** Bands represent the 95% Confidence Interval. For clarity, graphs display data up to day 600 from the outbreak onset. Decreases in survival probabilities occurred beyond day 600 are depicted as vertically dashed lines. These lines do not indicate sudden drops at that point but summarize posterior mortality up to 2.8 years.
(PDF)

**S5 Fig. Sub-sampling of prevalence of SCTLD by species at 100, 200, 300, 400, 500 and 600 days from the onset of epizootic.** Colors show the expected susceptibility group: High (red), Intermediate (yellow), and Low (blue). Panels show that prevalence comparisons between species differed at each time point. Furthermore, the prevalence levels of species mostly failed to match the expected susceptibility patterns across the sub-sampled periods.
(PDF)

**S1 Table. Frequency of use of parameters to classify species susceptibility to SCTLD.**
(PDF)

**S2 Table. Web of Science studies about coral epizootics from 1995 to 2025.**
(XLSX)

**S3 Table. GAMM's models validation.** The table shows the results of the normality, homoscedasticity, and autocorrelation tests on the residuals, as well as the model fit ($R^2$ values).
(XLSX)

**S4 Table. Onset and complete mortality days calculation through average tissue loss rates.**
(XLSX)

**S5 Table. Mean time of change from apparently healthy tissue adjacent to any lesion to each stage of colonization.** Status codes: Recently dead, denotes recently denuded skeleton; Film 1 is a scarce microbial/algal film; Film 2 is a well-stablished biofilm; Film 3 incipient turf algae mats; Turf Algae, denser turf algae mats; Macro Algae, macroalgal patches identifiable to genus level. (n) is the number of colonies analyzed to time calculations. CI = Confidence Interval of the 95%.
(PDF)

**S1 Dataset. Complete dataset. Containing dates and times of onset and mortality or recovery, size, type of lesion progression, and expected susceptibility of each colony.**
(XLSX)

## Acknowledgments

We thank the Posgrado en Ciencias del Mar y Limnología at UNAM for academic support leading to the doctoral degree of EOG-U. We acknowledge J. Rivera Ortega and R. Rodríguez Martínez for their invaluable assistance with monitoring; to E. Escalante Mancera and M. A. Gómez Reali from Meteorological and Oceanographic Monitoring Academic Service in Puerto Morelos for providing SST data; to E. Pérez Cervantes for technical support in BARCO lab; and F. Negrete Soto for his support with boat operations and diving logistics.

## Author contributions

**Conceptualization:** Edgar Omar Guzmán-Urieta.

**Data curation:** Edgar Omar Guzmán-Urieta.

**Formal analysis:** Edgar Omar Guzmán-Urieta.

**Funding acquisition:** Eric Jordán-Dahlgren, Lorenzo Alvarez-Filip.

**Investigation:** Edgar Omar Guzmán-Urieta.

**Methodology:** Edgar Omar Guzmán-Urieta , Eric Jordán-Dahlgren.

**Resources:** Lorenzo Alvarez-Filip.

**Software:** Edgar Omar Guzmán-Urieta.

**Supervision:** Eric Jordán-Dahlgren, Lorenzo Alvarez-Filip.

**Validation:** Edgar Omar Guzmán-Urieta , Eric Jordán-Dahlgren, Lorenzo Alvarez-Filip.

**Visualization:** Edgar Omar Guzmán-Urieta.

**Writing – original draft:** Edgar Omar Guzmán-Urieta.

**Writing – review & editing:** Edgar Omar Guzmán-Urieta , Eric Jordán-Dahlgren, Lorenzo Alvarez-Filip.

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
