## [Decision Letter · Decision Letter 0]

10 Sep 2025

Dear Dr. Guzman,

Thank you for submitting your manuscript to PLOS ONE. After careful consideration, we feel that it has merit but does not fully meet PLOS ONE’s publication criteria as it currently stands. Therefore, we invite you to submit a revised version of the manuscript that addresses the points raised during the review process.

We look forward to receiving your revised manuscript.

Kind regards,

Parviz Tavakoli-Kolour

Academic Editor

PLOS ONE

Journal Requirements:

 “The Institute of Marine Sciences and Limnology (ICMyL) and the National Autonomous University of Mexico (UNAM) providing financial and logistic support to EJ-D; Mexican Council of Science and Technology (CONACYT) grant the doctoral scholarship granted to EG-U (895565), and a research grant awarded to LA-F (FORDECYT-PRONACES/425888/2020).

ICMyL: https://www.icmyl.unam.mx/

UNAM: https://www.unam.mx/

CONACYT: https://secihti.mx/becas_posgrados/” 

3. We note that there is identifying data in the Supporting Information file <S2 Table>. Due to the inclusion of these potentially identifying data, we have removed this file from your file inventory. Prior to sharing human research participant data, authors should consult with an ethics committee to ensure data are shared in accordance with participant consent and all applicable local laws.

-Location data

Reviewers' comments:

Reviewer's Responses to Questions

**Comments to the Author**

1. Is the manuscript technically sound, and do the data support the conclusions?

Reviewer #1: No

Reviewer #2: Yes

Reviewer #3: Partly

2. Has the statistical analysis been performed appropriately and rigorously?

Reviewer #1: No

Reviewer #2: Yes

Reviewer #3: Yes

3. Have the authors made all data underlying the findings in their manuscript fully available?

Reviewer #1: Yes

Reviewer #2: Yes

Reviewer #3: Yes

4. Is the manuscript presented in an intelligible fashion and written in standard English?

Reviewer #1: Yes

Reviewer #2: Yes

Reviewer #3: Yes

Reviewer #1: Epidemiological analysis reveals coral species affected by Stony Coral Tissue Loss Disease present a similar epizootic progression despite differences in susceptibility and population impact. by E. Omar Guzmán-Urieta1, 2*, Eric Jordán-Dahlgren3, Lorenzo Alvarez-Filip2 PONE-D-25-35026

Overview: Authors attempt to describe temporal aspects of SCTLD in Mexican reefs. I like what authors are trying to do, but their approach seems overly convoluted and difficult to understand. Unless we have more details on how they did their surveys, estimates of underlying populations at risk, it becomes difficult to sort out potential confounding factors (like survey effort or number of animals available to observe) that could sway the results. The presentation of material is also garbled and does not make sense at least to this reviewer. There is definitely something interesting here but authors need to clarify, condense, and simplify to get their point across. I sense that the main points of the paper could be captured with Figures 2 and 4 with everything else deleted to simplify. Bottom line, you could get a good idea of interspecies susceptibility by plotting epidemic curves and calculating case fatality rate (the higher, the more susceptible the species). This would be cleaner and simpler to present to get your point across without a lot of convolution.

Line 79: Punctual prevalence is redundant. Just state prevalence which is percent of diseased animals at a point in time. Incidence is change in prevalence over time.

Lines 92-97: Difficult here to understand what you are saying. I suspect something along the following?: "Illustrative of this concept is the study by Dahlgren et al. [24] showing changes in prevalence of SCTLD by species over time."

Lines 104-105. Delete. It is obvious that a single sampling will not capture changes over time.

LInes 158-163: More detail on survey methodology would help here to assess validity of finding. A key aspect of temporal disease studies is implementing consistent effort spatially. Were permanent transects used and surveyed over time? Did you mark colonies? As written, it is difficult to say whether, for example, increased prevalence for a particular species was due to surveys with more of that species at a particular time point. It seems all this could be greatly simplified to present disease incidence by species to give a reader a sense of how disease varies between hosts coupled with case fatality rate for those colonies for which you have temporal observations. The latter would also address just how accurate Table 1 is for your region.

Lines 164-166: Were these colonies marked and followed over time? How were individual colonies recognized over time?

Lines 190-196: I don't understand. Fig. S1 is implying each colony has 2 observations but you are talking about 1 monitoring date.

Fiure S1. This caption and figure makes no sense to this reviewer. I just don't understand how you can calculate rate of tissue loss over time from a single observation.

Lines 197-205: This reads like Greek to me.

Figure S2: Seems all this could be simplified to something like acute, subacute and chronic tissue loss. I have a hard time distinguishing all the other categories you are stating

Lines 169-209: Seems all this could be considerably simplified and condensed to the following: Calculation of tissue loss rate over 2 or 3 time points using linear models. Attempting to calculate tissue loss rate from single observations is stretching it and would suggest delete those two parts (lines 197-205).

Line 215: Do you mean: "These curves were computed by species and by species grouped to expected susceptibility (high,intermediate and low) (Cite)"?

Lines 216-220: What exactly are you trying to say here?

Lines 227-229: condense: "Epi curves were generated from weekly tallies."

Lines 236-239: What is this association you are trying to make? How exactly did you compare density distribution of disease vs mortality? And why only compare density of disease between sites but nor mortality?

Line 262: Earlier, you said epidemic curves were generated from weekly tallies. Now you say daily. Which is it?

Lines 287-289: Delete. This does not add anything meaningful to previous sentence and overly complicates things and seems arbitrary (what is magical about 5%? why not 10%? 20%?).

Lines 291-298: Just how meaningful is this? Can you delete to simplify and get to the point?

Lines 299-326: Ditto.

Figure 2. Here it would be helpful to have actual N of colonies represented for each plot. I suspect for some plots (e.g. meandrina, dichocoenia,Porites), those are simply reflective of fact that those species are likely less numerous on reefs. Indeed, consider plotting curves only for species for which you have a minimal N (e.g. 20+) to make the data meaningful. Indeed, based on your data, I suspect susceptibility Table 1 is simply a function of how commonly those species are encountered and this could be a confounding factor (more disease simply because it is easier to detect and may not have anything to do with species susceptibility). Any way to correct for that? Indeed, a plot between relative numbers of each species at a site and the slope of population at risk (blue line) in your plot figure 2 might be informative. The closer the slope is to 0 or 1 suggests rare species which would be only those corals in intermediate slope values should really be considered when assessing epidemic curves.

Figure 2 caption: "This is the point of time when the number of newly diseased individuals was maximum and then began to decline." Delete...redundant. Flashing line...do you mean solid black line?

Figure 3. Consider deleting. This really does not add anything meaningful to your story and is redundant to Figure 2.

Lines 404-407:"Furthermore, the species A. agaricites, A. tenuifolia, I. sinuous, S. intersepta, and P. porites exhibited low survival probabilities for their diseased populations (below 11 %), despite their at-risk populations showing moderate to high survival probabilities (S4 Fig)

Figure 5 and 6: Delete. Does not really add anything meaningful to the story.

Keep figure S2 and S3 and delete the rest. Those do not add anything meaningful to story.

Lines 418-421: Delete. Not sure what this adds.What pairwise comparisons were different?

Lines 422-429: Condense to: "there were no statistically significant differences in

429 survival probabilities among the diseased populations."

Lines 432-438: What is your point here? This paragraph makes no sense.

Lines 439-444: Seems from FIgure 2 that prevalence peaked out at 1 for all species suggesting they are all equally susceptible no?

Lines 446-492: This is all very convoluted, and without an appreciation of underlying N for each species surveyed, it is difficult to sort out here whether these data are indicative of species susceptibility or simply an artefact that there is less disease evidence in species that are less visible or common.

Line 457/Fig 6: Why are you splitting them on PC1? Why not PC2? Also, not understanding the rationale for the cutoffs for the 4 groups on PC1? Seems I could make my on grouping 1-4 based on PC quadrats. For example for outbreak length (PSTR, SSID, PCLI, MCAV), P survival (PAST, AAGA< OFAV, ATEN, OANN), etc. This all seems arbitrary to me.

Lines 510-512: It is already known that there is difference in species susceptibility to SCTLD.

Reviewer #2: Please see attached file.

Reviewer #3: I recommend that this manuscript needs significant revisions. This study addresses an important topic by applying epidemiological tools to coral disease; however, there are concerns regarding the limitations of the models and the results not supporting the claims in the discussion that need to be addressed before publication. Please see below for more comments and line by line suggestions.

PONE-D-25-35026 Comments to Authors

General comments:

The conclusion that mortality appears to contribute to secondary transmission events should be more of a hypothesis as it is speculative. I suggest changing the terminology throughout the manuscript to soften the language as this is a hypothesis.

The figure descriptions are very long and also have some discussion points in them. I suggest removing the discussion points from the legends and including them in the manuscript if necessary (e.g., lines 104-105 are discussion points). Some of the figures are complicated, so potential reworking may be needed.

Point prevalence is the more commonly used term compared to punctual prevalence and should be changed throughout the manuscript.

The limitations of the models need to be highlighted and discussed throughout the manuscript.

General methods:

Though there was no difference in spatial distribution between sites, there needs to be evidence presented (I.e. supplementary statistics) that there are no incidence differences between sites. Since the authors are referring to temporal dynamics and not spatial distribution for these data, analyses need to be run to determine if the sites can be pooled.

Why were only 422 corals used to estimate tissue loss per day when 990 had signs of SCTLD? It says further analyses were used for these 990, but more than half were not put into any of the models. Was a model applied to the other >500 corals? If so, which one was it applied to? That needs to be stated directly. If not used in this study, then those should be removed from the methods, table 1, discussion, etc.

Models 3 and 4 – These need to be presented with more caution. Model 3 relies on observations of corals of the same species and size while Model 4 relies on a perceived timeline, and it needs to be addressed that these are less reliable methods to infer incidence.

For the PCA – how were the variables scaled?

Species with a low sample size need to be addressed as a limitation of these models. For example, M. jacksoni is presented within figure 2, but only had a sample size of 2.

Figure 2 is overly complex, making it difficult to interpret. Consider simplifying or splitting into multiple panels.

General discussion:

The structure of the discussion needs to be re-worked to have a better flow. Suggestion: summary of main findings, comparisons with previous work, interpretations of key patterns, limitations, implications for future use.

The language has to be softened within this section. For example, “point prevalence may not always be a reliable indicator of susceptibility, especially when measured at a single time point.”

The PCA provides an exploratory framework for susceptibility, and it complements existing classifications, but tone down the language regarding the novel classification.

In general, the language regarding the “secondary waves caused by sloughed off tissue” needs to be softened as it is a hypothesis and cannot be proven with modeling.

It is unusual to have phrased questions throughout – consider revising for easier reading.

Lines 636-650 – it is unclear why there is significant detail regarding opportunistic infections and the coral’s response. I suggest cutting down on these sections as they have been described in other studies and do not necessarily discuss the results of this study. Or potentially adjust the flow of the discussion so that the interpretation of susceptibility differing is clearer.

I highly suggest adding a limitations section to the discussion where the drawbacks of the models are discussed. For example, limitations include variation in sample size, inferred incidence for Models 3 and 4, pooling sites, non-independence of colonies, etc.

Consider adding a final paragraph discussing the implications of the modeling and how they can be used in the future. Suggestions include for future outbreaks, provide targeted interventions, aid mangagement decisions, etc.

Finally, consider adding Aeby et al 2025 “Progressive chronic tissue loss disease in Siderastrea siderea on Florida’s coral reef” as a potential explanation for the Sids species susceptibility differences.

Line by line suggestions:

Line 38 – mortality may contribute to secondary transmission

Line 41 – say what you are comparing the higher level of susceptibility to – i.e., previous studies

Line 51 – sensible doesn’t fit here. Sensitive? Susceptible?

Line 58 – stony coral tissue loss disease is lowercase

Lines 73 – 76 – the wording is confusing to read. Revise the “will also have likely compromised.” Potentially break into two sentences as well for clarity

Line 79 – Change “punctual prevalence” to “point prevalence” throughout the manuscript as it is more the more commonly used term.

Line 84 – change “lesions apparition” to “lesion appearance” if that maintains the meaning of the sentence

Line 88 - “giving an incomplete picture” is awkward phrasing for the sentence. Consider revising

Line 100 figure 1 description – there are no white arrows within the figure, so assuming you are referring to black arrows. Remove discussion/result points from the figure descriptions.

Line 111 – Revise sentence to avoid starting with “which.” Potentially combine with previous sentence.

Line 137 – Consider “epi-curves, Kaplan-Meier risk curves, and survival curves” if it maintains meaning.

Line 182 – Repeated measures within the same individuals may violate independence assumptions of some models and should be stated that it should be interpreted with caution.

Line 219 – change to epizootic

Lines 233-236 – the language here needs to be softened as correlation does not mean causation, and this is a hypothesis

Line 270-273 – revise sentence for readability

Line 289 – change “umbral” to “threshold”

Line 292 – change to “on the other hand”

Line 339 figure 2 description – “flashing is unclear” changed to dashed or change the figure so that the dotted and dashed black lines are easier to differentiate

Lines 357 – 371 – the sample size of the corals needs to be clearly addressed when comparing between species.

Line 381-382 – revise for clarity

Lines 392-393 – revise to “this patten is consistent with the hypothesis” as it does not prove it. It is not possible to confirm that sloughing off tissue is causing a secondary wave with these models alone.

Line 409 figure 5 description – revise for clarity about B and C

Line 429 – change to log rank

Line 471 – revise for clarity

Figure 6 and description – the four “levels” of susceptibility are not based on formal clustering, which makes this classification subjective. Reframe to be exploratory instead of discrete groups. Clarify how the variables were standardized before the PCA. The legend is dense and hard to follow, so the text needs to be simplified. The results are interesting, but the interpretation should be more cautious and should be presented as exploratory rather than definitive evidence of a new susceptibility classification.

Lines 510-512 – revise for clarity

Line 512 – species' if it maintains clarity

Line 527 – point prevalence may not be a reliable measure

Line 557 – outbreak duration may not be a reliable measure

Lines 574-575 – revise for clarity

Lines 581 – 582 – soften this language as this is not direct evidence from the field and these observations need experimental manipulations instead of modeling to prove causation and not just correlation

Line 608 – revise so the sentence does not start with which

Lines 610-619 – add Palacaio-Castro “elevated temperature decreases stony coral tissue loss disease transmission, with little effect of nutrients” to discussion

Line 652 - “is not against” is unclear to read

Line 671 – misused may not be the appropriate term here

Line 672-674 – soften the language here as the models could aid in elucidating the etiology of SCTLD

**Do you want your identity to be public for this peer review?** For information about this choice, including consent withdrawal, please see our Privacy Policy

Reviewer #1: **Yes: ** Thierry M. Work

Reviewer #2: **Yes: ** Aine Hawthorn

Reviewer #3: No

---

## [Author Response · Author response to Decision Letter 1]

5 Nov 2025

Response to Academic Editor:

Academic Editor recommendations:

We have reviewed the PLOS ONE style requirements and ensured that they are followed in the manuscript, including file naming.

2. Please provide an amended statement that declares *all* the funding or sources of support (whether external or internal to your organization) received during this study, as detailed online in our guide for authors at http://journals.plos.org/plosone/s/submit-now. Please also include the statement “There was no additional external funding received for this study.” in your updated Funding Statement. Please include your amended Funding Statement within your cover letter. We will change the online submission form on your behalf.

I have included the founding statement at the end of the cover letter, including the requested information.

3. We note that there is identifying data in the Supporting Information file <S2 Table>. Due to the inclusion of these potentially identifying data, we have removed this file from your file inventory. Prior to sharing human research participant data, authors should consult with an ethics committee to ensure data are shared in accordance with participant consent and all applicable local laws.

I appreciate the comment; however, it appears to be a misunderstanding. Table S2 contains a bibliographic summary of scientific publications related to coral disease research. It includes only the information necessary for citation purposes, such as authors' names, publication title, year, journal name, and similar details. This work does not involve human subjects.

Thank you very much. We have considered including additional citations where appropriate.

Response to Reviewers:

Reviewer #1

Overview: Authors attempt to describe temporal aspects of SCTLD in Mexican reefs. I like what authors are trying to do, but their approach seems overly convoluted and difficult to understand. Unless we have more details on how they did their surveys, estimates of underlying populations at risk, it becomes difficult to sort out potential confounding factors (like survey effort or number of animals available to observe) that could sway the results. The presentation of material is also garbled and does not make sense at least to this reviewer. There is definitely something interesting here but authors need to clarify, condense, and simplify to get their point across. I sense that the main points of the paper could be captured with Figures 2 and 4 with everything else deleted to simplify. Bottom line, you could get a good idea of interspecies susceptibility by plotting epidemic curves and calculating case fatality rate (the higher, the more susceptible the species). This would be cleaner and simpler to present to get your point across without a lot of convolution.

We sincerely thank the reviewer for their thoughtful and constructive comments. In response, we have carefully revised and substantially edited the Methods section to clarify the survey design, sampling effort, and estimation of populations at risk. We also reviewed the entire manuscript to improve clarity and flow, aiming to make our approach more accessible. Please refer to your and the other reviewer's specific comments for details on how we have amended the manuscript.

We appreciate the suggestion to focus on Figures 2 and 4, which indeed highlight key aspects of susceptibility at population and community levels. However, we consider the additional analyses essential to support a more robust and biologically grounded interpretation. In coral disease studies, susceptibility is often inferred from prevalence alone, which overlooks the temporal progression of outbreaks, survival dynamics, and epidemic risk. Our approach, based on epidemic curves and Kaplan-Meier survival analysis, was designed to address these limitations and evaluate susceptibility in a temporally explicit framework.

Although this may increase the apparent complexity of the work, we consider it necessary to reflect the biological and epidemiological realities of SCTLD. While we acknowledge the communicative value of case fatality rate, we believe our framework already offers a more comprehensive representation of disease severity and susceptibility without calculating CFR separately.

Line 79: Punctual prevalence is redundant. Just state prevalence which is percent of diseased animals at a point in time. Incidence is change in prevalence over time.

Thank you. We agree and have changed to ‘prevalence’ as suggested.

Lines 92-97: Difficult here to understand what you are saying. I suspect something along the following?: "Illustrative of this concept is the study by Dahlgren et al. [24] showing changes in prevalence of SCTLD by species over time."

Thank you. We have reworded this section (lines 90 to 96) to enhance clarity and better convey the intended argument.

Lines 104-105. Delete. It is obvious that a single sampling will not capture changes over time.

Deleted as suggested.

LInes 158-163: More detail on survey methodology would help here to assess validity of finding. A key aspect of temporal disease studies is implementing consistent effort spatially. Were permanent transects used and surveyed over time? Did you mark colonies? As written, it is difficult to say whether, for example, increased prevalence for a particular species was due to surveys with more of that species at a particular time point. It seems all this could be greatly simplified to present disease incidence by species to give a reader a sense of how disease varies between hosts coupled with case fatality rate for those colonies for which you have temporal observations. The latter would also address just how accurate Table 1 is for your region.

Lines 164-166: Were these colonies marked and followed over time? How were individual colonies recognized over time?

The two previous comments will be answered below.

We thank the reviewer for these valuable observations. We have expanded the methods section to explain these key aspects of data collection. This more detailed description is between lines 155 – 170, as follows:

“At Bocana, four sampling points were established: B1 (453 m²), B2 (19.5 m²), B3 (140 m2), and B4 (60 m²). At Picudas, three points were defined: P1 (435 m²), P2 (12 m²) and P3 (9 m²). These polygons were delimited by reef topography and visibility constraints, but remained fixed throughout the study. Despite differences in size, consistent effort was applied within each polygon to document all visible coral colonies across species during each visit through photographic surveys.

Initially, monitoring was focused on spatial disease patterns in Pseudodiploria strigosa, as part of a previous study on SCTLD dynamics [26]. However, during those surveys, photographic records were systematically taken of all visible coral colonies within the same sampling areas, regardless of species. Each colony was photographed individually during every monitoring event, with multiple angles captured depending on its size and structural complexity. These images allowed us to track colonies over time and retrospectively assess their health status (healthy, diseased, or dead), visually estimate the percentage of dead tissue caused by SCTLD or previous mortality, and classify lesion progression type (acute, subacute, or chronic, sensu Work & Aeby [46]). Additionally, the maximum diameter of each colony at the onset of the study was measured using ImageJ (1.52a), based on scale markers included in the photographs.”

Lines 190-196: I don't understand. Fig. S1 is implying each colony has 2 observations but you are talking about 1 monitoring date.

Fiure S1. This caption and figure makes no sense to this reviewer. I just don't understand how you can calculate rate of tissue loss over time from a single observation.

Lines 197-205: This reads like Greek to me.

The three previous comments will be answered below.

Thank you for your comments. This part of the method aims to estimate the onset of lesion signs and the timing of total tissue loss in colonies with only one recorded observation. To improve clarity, we have revised the manuscript text and restructured it as follows (now in lines 209 – 238):

“For colonies with only one recorded observation of tissue loss, we acknowledge that direct estimation of lesion progression is not possible. To avoid excluding these cases, we implemented two approaches based on biologically informed assumptions and empirical reference data. These methods aim to reconstruct plausible trajectories while accounting for uncertainty, as detailed in points III and IV.

III. The third approach was implemented for diseased colonies with only one recorded instance of tissue loss. These cases occurred either because the colonies were too small to relocate or because they died and were likely rapidly overgrown by algae, resulting in their capture as diseased during only one survey (n = 130). To address this limitation, we assumed that these individuals followed a similar trajectory to others of the same species, size category (small: <33 cm; medium: >33 to 66 cm; large: >66 to <100 cm; and extra-large: >100 cm), and lesion progression type. Based on this assumption, we first calculated individual tissue loss trajectories from colonies with sufficient temporal data, grouped by species, size category, and lesion progression type. From these trajectories, we derived an average daily tissue loss rate, which was then used to interpolate the percentage of tissue loss for colonies with only one recorded observation. This allowed us to estimate the full progression trajectory, including onset and extent of tissue loss, for single-observation cases. This procedure is illustrated in S1 Fig. A 95% confidence interval for each extrapolated trajectory was computed using the bootstrap percentile method and is provided for reference (S4 Table).

IV. For diseased colonies with only one tissue mortality record that could not be matched to any of the above categories of species, size, and lesion progression (n = 88), we estimated their individual tissue loss trajectories based on the colonization stage of the exposed skeleton. These stages (recent mortality, algal/microbial film, turf algae, and macroalgae) follow a consistent post-mortality succession (S2 Fig). Each stage was associated with an average transition time, calculated from a reference subsample of 157 colonies (S5 Table). Using this information, we inferred when each skeletal region was covered by living tissue. Colonies displaying multiple colonization stages provided several time points along the lesion progression. Depending on the number of inferred points, we then applied either a GAMM or a linear interpolation method (as in cases I and II, respectively) to reconstruct the full trajectory of tissue loss, from lesion onset to complete mortality.”

And the S1 Figure legend accordingly (lines 944 – 951):

“S1 Fig. Estimation of lesion progression for colonies with only one recorded tissue loss value. Example for P. strigosa (medium size, acute lesion progression, n = 12): A) Daily tissue loss trajectories from colonies with multiple observations were grouped by species, size category, and lesion progression type. Each colored line is an individual trajectory. B) Time was normalized around the 50% tissue loss point to align trajectories and estimate average trajectory. C) The average trajectory (blue line) and its 95% confidence interval (blue ribbon) were computed via bootstrap percentile method. D) Red dots show single-record colonies; their lesion progression was extrapolated using the reference trajectory, assuming similar dynamics.”

Figure S2: Seems all this could be simplified to something like acute, subacute and chronic tissue loss. I have a hard time distinguishing all the other categories you are stating

Thank you very much for your comment. In this case, the categories correspond to colonization stages. Although there is a close relationship between these colonization stages and the rate of tissue loss, we want to relate time to the cover sequence. Therefore, we have retained the category structure explained in the legend for Figure S2 but revised both the legend and the figure itself to improve clarity and visual interpretation.

Lines 169-209: Seems all this could be considerably simplified and condensed to the following: Calculation of tissue loss rate over 2 or 3 time points using linear models. Attempting to calculate tissue loss rate from single observations is stretching it and would suggest delete those two parts (lines 197-205).

Thank you for your comment. We understand the concern regarding inference from single observations. However, our goal was not to calculate tissue loss rates directly from isolated data points, but rather to reconstruct plausible progression trajectories using systematic, biologically informed methods. For colonies with only one recorded observation, we relied on patterns derived from individuals of the same species, size category, and lesion type (approach III), or from the colonization stage of the exposed skeleton (approach IV), both of which are grounded in empirical data. These methods allowed us to estimate onset and extent of tissue loss in a reproducible and transparent way, while acknowledging uncertainty through confidence intervals (S4 & S5 tables). We believe this approach adds value by recovering otherwise lost information and enabling inclusion of these cases in broader epidemiological analyses. Nonetheless, we have revised the text to improve clarity and emphasize the assumptions and limitations involved. The clarified text is now found between lines 180 and 242, cited above. On the other hand, we have added a note of methodological considerations and limitations at the end of the discussion where we include this point (lines 649 – 656):

“For colonies with only one recorded observation of tissue loss, inference-based approaches (III and IV, see methods) were applied transparently, incorporating ecological indicators and uncertainty. Among these single-record cases, the distribution across susceptibility groups was approximately balanced (57% high, 44.5% intermediate, 62% low; relative to diseased colonies in each group), suggesting no systematic bias toward any group. A more influential factor was colony size: 61% of these cases corresponded to small colonies. We consider that this approach adds value by recovering information that would otherwise be excluded and by enabling the inclusion of these cases in broader epidemiological analyses”

Line 215: Do you mean: "These curves were computed by species and by species grouped to expected susceptibility (high,intermediate and low) (Cite)"?

Exactly, we have revisited this section to improve clarity and ensure the grouping logic is clearly conveyed. Lines 247-253.

Lines 216-220: What exactly are you trying to say here?

We have adjusted the text, now on lines 249-253 for clarity:

“This analysis evaluates whether species grouped by expected susceptibility-based groups exhibit consistent differences not only in period prevalence and cumulative mortality, but also in other epidemiological indicators throughout the course of the epizootic. The list of species by group is provided in Table 1.”

Lines 227-229: condense: "Epi curves were generated from weekly tallies."

We appreciate the suggestion and have condensed, now in lines 265-266.

Lines 236-239: What is this association you are trying to make? How exactly did you compare density distribution of disease vs mortality? And why only compare density of disease between sites but nor mortality?

Thank you for your observations. We have clarified the text to specify that the association being explored is whether mortality could help explain the observed incidence patterns. Additionally, we clarified that both incidence and mortality distributions were co

---

## [Decision Letter · Decision Letter 1]

1 Dec 2025

Epidemiological analysis reveals coral species affected by Stony Coral Tissue Loss Disease present a similar epizootic progression despite differences in susceptibility and population impact.

PONE-D-25-35026R1

Dear Dr. Guzman,

We’re pleased to inform you that your manuscript has been judged scientifically suitable for publication and will be formally accepted for publication once it meets all outstanding technical requirements.

Kind regards,

Parviz Tavakoli-Kolour

Academic Editor

PLOS ONE

Additional Editor Comments (optional):

Reviewers' comments:

Reviewer's Responses to Questions

**Comments to the Author**

Reviewer #2: All comments have been addressed

2. Is the manuscript technically sound, and do the data support the conclusions?

Reviewer #2: Yes

3. Has the statistical analysis been performed appropriately and rigorously?

Reviewer #2: Yes

4. Have the authors made all data underlying the findings in their manuscript fully available?

Reviewer #2: Yes

5. Is the manuscript presented in an intelligible fashion and written in standard English?

Reviewer #2: Yes

Reviewer #2: (No Response)

**Do you want your identity to be public for this peer review?** For information about this choice, including consent withdrawal, please see our Privacy Policy

Reviewer #2: **Yes: ** Aine Hawthorn

---

## [Editor Report · Acceptance letter]

PONE-D-25-35026R1

PLOS One

Dear Dr. Guzmán-Urieta ,

I'm pleased to inform you that your manuscript has been deemed suitable for publication in PLOS One. Congratulations! Your manuscript is now being handed over to our production team.

Kind regards,

on behalf of

Dr. Parviz Tavakoli-Kolour

Academic Editor

PLOS One